# HER2DX in older patients with HER2-positive early breast cancer: extended follow-up from the RESPECT trial of trastuzumab ± chemotherapy

Kazuki Nozawa[1,2], Masataka Sawaki[3], Yukari Uemura[4], Michiko Tsuneizumi[5], Toshimi Takano [6], Naomi Gondo[7], Fumikata Hara[8], Michiko Harao[9], Tatsuya Toyama [2], Naruto Taira[10], Ana Vivancos [11], Charles M. Perou[12], Esther Sanfeliu[11], Fara Brasó-Maristany[11], Joel S. Parker[11], Wesley Buckingham [11], Laia Paré[11], Guillermo Villacampa[13], Mercedes Marín-Aguilera[11], Patricia Villagrasa[11], Aleix Prat [11,14,15,16,17,18] ✉ & Hiroji Iwata[1,18]

Older adults with HER2-positive early breast cancer are underrepresented in clinical trials, and the benefit of chemotherapy in this population remains uncertain. We evaluated the HER2DX genomic assay within the randomized RESPECT trial (NCT01104935), which compared adjuvant trastuzumab with or without chemotherapy in patients aged 70–80 years. In this prespecified translational analysis (Trans-RESPECT), HER2DX scores were available for 154 patients. The HER2DX risk score classified 74.0% as low risk and 26.0% as high risk. Ten-year relapse-free and overall survival were higher in the low-risk group. HER2DX remained independently associated with overall survival in multivariable analysis. The HER2DX immune, luminal, and proliferation signatures that compose the risk score were also prognostic. While the HER2DX pCR score was not prognostic overall, exploratory subgroup analyses suggested a potential survival benefit from chemotherapy in the pCR-high group. HER2DX offers prognostic value and may guide chemotherapy use in older patients with HER2-positive early breast cancer. **Clinical Trial Information** NCT01104935

The treatment landscape for early-stage HER2-positive (HER2+) breast cancer has evolved considerably over the past two decades[1,2]. While the combination of trastuzumab and chemotherapy remains the cornerstone of adjuvant therapy in most patients, this approach can be associated with significant toxicity, particularly in older adults[3–6]. As the global population ages, the number of patients aged 70 years and older diagnosed with HER2+ early breast cancer continues to grow[7], yet this group remains underrepresented in pivotal clinical trials[8]. Consequently, there is a critical need to identify strategies that preserve efficacy while minimizing treatment-related adverse effects in this population.

The RESPECT trial was a randomized phase III study conducted in Japan that directly addressed this issue by comparing trastuzumab monotherapy to trastuzumab plus chemotherapy in patients aged 70–80 years with surgically resected HER2+ early breast cancer[9]. Although the trial did not meet its primary endpoint of demonstrating noninferiority of trastuzumab alone, the observed absolute difference in survival was modest, with a < 1-month loss in restricted mean

survival time at 3 years, and trastuzumab monotherapy was associated with substantially fewer grade 3–4 adverse events and improved health-related quality of life[9]. These findings highlighted the potential of de-escalated regimens in selected older patients but also underscored the need for robust biomarkers to guide such decisions.

HER2DX is a multigene expression assay specifically designed for early-stage HER2+ breast cancer[10–21]. It integrates tumor-intrinsic biology with clinical features to provide a genomic risk score (i.e., HER2DX risk-score) for long-term outcomes, as well as additional biologically driven metrics such as a pathologic complete response (pCR) likelihood score (i.e., HER2DX pCR-score) and an ERBB2 expression score[15,18]. The assay has been validated in several clinical trial cohorts, including studies using less intensive therapy such as APT[17,22,23], and ATEMPT[16] trials, and has demonstrated consistent prognostic and predictive performance. A recent prospective real-world study confirmed its clinical impact, showing that HER2DX results led to treatment adjustments in nearly half of patients, most commonly through reduction in treatment intensity[24]. HER2DX is used to guide decision-making across various clinical scenarios, including chemotherapy intensity, use of neoadjuvant therapy, and HER2 status confirmation. The test is currently incorporated into the Spanish breast cancer guidelines[25] and was discussed at the 19th St. Gallen Breast Cancer Conference in 2025[26]. However, its utility in older patients, particularly those receiving trastuzumab monotherapy, has not been previously explored.

To address this knowledge gap, we conducted the Trans-RESPECT study, a prospective-retrospective analysis of tumor samples from the RESPECT trial[9], to evaluate the prognostic and potential predictive value of the HER2DX assay in patients aged 70–80 years treated with adjuvant trastuzumab with or without chemotherapy. Specifically, we investigated whether HER2DX could identify patients with such excellent long-term outcomes that the addition of chemotherapy is unlikely to demonstrate a survival benefit and therefore may safely forego chemotherapy. The HER2DX risk score and pCR score were also evaluated for their ability to identify a group of patients who experienced a survival benefit from the addition of chemotherapy in this older population.

## Patients and methods

### Study design and participants
This was a prespecified translational study (Trans-RESPECT) using tumor samples and clinical data from the RESPECT trial (NCT01104935), a phase III randomized study conducted by the Comprehensive Support Project for Oncology Research (CSPOR)[9]. The primary prespecified objective of Trans-RESPECT was to assess whether relapse-free survival (RFS) differed significantly between patients classified as low versus high risk by the HER2DX risk score. Overall survival (OS) was a key secondary endpoint. Additional prespecified analyses included testing whether the HER2DX pathologic complete response (pCR) likelihood score predicted RFS or OS benefit from the addition of chemotherapy.

RESPECT enrolled women aged 70–80 years with stage I–IIIA HER2+ early breast cancer who had undergone definitive surgery. Patients were randomly assigned to receive adjuvant trastuzumab monotherapy or trastuzumab plus chemotherapy. Patients with hormone receptor–positive disease received standard adjuvant endocrine therapy. The trial included 275 patients with a mean age of 73.5 years and a mean follow-up of 4.1 years. The primary endpoint of non-inferiority for trastuzumab monotherapy was not met; however, the difference in 3-year restricted mean survival time between arms was less than 1 month. The 3-year RFS was 92.4% (95% CI, 86.3–95.8) with trastuzumab monotherapy versus 95.3% (89.7–97.8) with trastuzumab plus chemotherapy. The 3-year OS was 97.2% (91.2–99.1) in the monotherapy group and 96.6% (89.5–98.9) in the combination group. The toxicity profile and health-related quality of life outcomes favored

trastuzumab monotherapy, supporting its potential use in selected older patients.

### HER2DX genomic assay
HER2DX testing was performed in a central laboratory located in Barcelona, Spain, under applicable In Vitro Diagnostic Regulation (IVDR) standards[13,15]. Only one sample per patient was profiled, corresponding to the primary tumor resected during surgery. The HER2DX test outputs a risk score and a pCR score which are derived from 4 predefined gene expression signatures comprising a total of 27 genes, capturing immune infiltration, tumor cell proliferation, luminal differentiation, and expression of the HER2 amplicon. The immune signature used was the 14-gene immunoglobulin (IGG) module (CD27, CD79A, HLA-C, IGJ, IGKC, IGL, IGLV3-25, IL2RG, CXCL8, LAX1, NTN3, PIM2, POU2AF1, and TNFRSF17), previously defined by unsupervised clustering of breast tumors and independently validated as prognostic in large datasets of untreated patients[27]. The other three signatures were derived from unsupervised clustering of 185 breast cancer–related genes in the HER2+ Short-HER training dataset[15]. Highly correlated gene clusters (Pearson r > 0.80) were used to define the remaining signatures: the proliferation signature (EXO1, ASPM, NEK2, KIF23), the luminal differentiation signature (BCL2, DNAJC12, AGR3, AFF3, ESR1), and the HER2 amplicon signature (ERBB2, GRB7, STARD3, TCAP)[15].

The HER2DX risk score was evaluated as a binary variable: low risk [1–49] and high risk [50–99]. The HER2DX risk score was computed using the validated model developed in the ShortHER trial[15]. In the ShortHER training dataset, the risk stratification cutoff was predefined to ensure that the low-risk group maintained >90% distant recurrence-free survival at 3, 5, and 7 years[15,18]. Patients classified as high-risk fall outside this threshold and have a comparatively higher risk of recurrence. The HER2DX pCR likelihood score was assessed as a categorical variable: pCR-low (1–32), pCR-medium (33–67), and pCR-high (68–99). The three biological signatures that compose the HER2DX risk score—immune/IGG, luminal, and proliferation—were also evaluated independently in an exploratory analysis. Cutoffs for each component signature were based on tertile distributions defined in the training dataset (i.e., Short-HER study[15]). All genomic analyses were performed by Reveal Genomics, blinded to clinical and outcome data until the genomic data were transferred to CSPOR for integrated analysis.

### Endpoints
The co-primary endpoints were RFS, defined as the time from randomization to the first invasive recurrence, second primary cancer, or death from any cause, and OS, defined as the time from randomization to death from any cause.

### Statistical analyses
Survival endpoints were estimated using the Kaplan–Meier method. Univariable and multivariable Cox proportional hazards models were used to estimate hazard ratios (HR) with 95% confidence intervals (95% CI). When evaluating the prognostic value of HER2DX scores (risk and pCR scores), treatment arm (trastuzumab alone vs. trastuzumab plus chemotherapy) was included as a stratification factor to account for potential differences in baseline hazard functions across treatment groups. This allowed for estimation of a common hazard ratio for HER2DX-defined subgroups while adjusting for treatment-related differences in baseline risk.

To evaluate the predictive value of HER2DX scores, interaction tests were performed using Cox models with a treatment-by-HER2DX interaction term, to assess whether the effect of chemotherapy on survival outcomes differed by HER2DX subgroup classification. Analyses were conducted for the HER2DX-evaluable population and for the node-negative subgroup. Sensitivity analyses were conducted using right-censoring at 5, 6, 7, 8, 9, and 10 years, given the potential

relevance of time-dependent effects in older patients. The proportional hazards assumption was tested and inspected visually using Schoenfeld residuals.

For data analysis, the HER2DX pCR group that was low and medium was pooled into a single category. No data imputation was performed. The median follow-up was calculated using the reverse Kaplan–Meier method. A significance level of 0.05 was set for a two-sided test, and all analyses were performed using R statistical software version 4.3.1. Pearson correlation coefficients (r) were used to assess linear associations between HER2DX signature scores, calculated using the cor() function in base R.

### Data access and ethical considerations
All individual patient data used in this study were accessed under a formal data-sharing agreement between CSPOR and Reveal Genomics. Data were pseudonymized at the source in accordance with applicable data protection regulations. All procedures conformed to institutional and international ethical standards, including the Declaration of Helsinki, and were conducted in compliance with national and international regulations governing human subject research and data protection.

### Role of the funding source
The study was designed and performed by investigators from CSPOR and Reveal Genomics. All authors had full access to the data and had final responsibility for the decision to submit the manuscript for publication.

## Results
### Patient characteristics
A total of 154 out of 266 eligible patients (58.0%) from the RESPECT trial had tumors successfully profiled with the HER2DX test and were included in this analysis. This subset reflects the 172 FFPE tumor blocks retrieved from local sites, of which 166 (96.5%) met the minimum tumor requirements and 163 (98.2%) met the minimum RNA yield/quality requirements. Overall, 154 samples (94.5%) were successfully profiled, with failures due to insufficient tumor tissue ($n = 6$) or insufficient RNA yield/quality ($n = 3$) (Supplementary Fig. 1). The overall profiling success rate from tumor block retrieval to HER2DX results was 89.5%. Baseline clinical and pathological characteristics of these patients are summarized in Table 1. No major differences were observed between the Trans-RESPECT subset and the overall RESPECT trial population. The mean age was 73.7 years, and 44.2% had estrogen receptor-positive disease. Compared with the full RESPECT cohort, the HER2DX-evaluable population was well balanced across arms and representative in terms of tumor stage and nodal status.

Among the 154 patients, 74 (48.1%) were assigned to trastuzumab monotherapy and 80 (51.9%) to trastuzumab plus chemotherapy. When stratified by HER2DX risk score, patients classified as high risk more frequently had node-positive disease (52.5% vs 3.5%) and larger tumors (pT2–3: 77.5% vs 38.6%) compared with those classified as low risk. The median follow-up was 9.3 years (95% CI 9.1–9.6). During this period, 30 RFS events and 22 OS events were observed. Notably, all OS events occurred after an RFS event: 124 patients experienced neither event, 8 had an RFS event without subsequent death, and 22 experienced both. No patients died without a preceding recurrence and, among the 22 patients who experienced both recurrence and death, 41% died at the time of their first recurrence, while the remainder survived for a median of 2.2 years following recurrence.

Consistent with the original RESPECT trial[9], no significant differences in RFS (HR = 0.79; 95% CI, 0.38–1.62; p-value = 0.52) or OS (HR = 0.73; 0.32–1.70; p-value = 0.47) were observed between treatment arms in the Trans-RESPECT population, despite extended follow-up (Supplementary Figs. 2, 3). In univariable analyses, tumor stage, nodal status, and hormone receptor status were not statistically significantly associated with RFS or OS in the overall population, while age was the only variable associated with survival outcomes (Supplementary Tables 1, 2). Among patients with node-negative disease, neither tumor stage nor hormone receptor status was prognostic (Supplementary Tables 3, 4).

### HER2DX risk score and prognostic value
The HER2DX risk score classified 114 tumors (74.0%) as low-risk and 40 tumors (26.0%) as high-risk. In the trastuzumab monotherapy arm, 78.4% were low risk and 21.6% high risk, while in the trastuzumab plus chemotherapy arm, 70.0% were low risk and 30.0% high risk. Patients in the HER2DX low-risk group had better long-term outcomes compared with those in the high-risk group (Fig. 1A–D). The 10-year RFS was 77.9% in the low-risk group versus 68.0% in the high-risk group (HR = 0.48; 95% CI 0.23–1.01; p-value = 0.05), and the 10-year OS was 85.9% versus 69.7%, respectively (HR = 0.34; 95% CI 0.15–0.80; p-value = 0.01) (Fig. 1A, B). In the node-negative subgroup, the 10-year RFS was 77.8% in low-risk patients versus 58.6% in high-risk patients (HR = 0.36; 95% CI 0.15–0.86), and the 10-year OS was 86.3% versus 47.7%, respectively (HR = 0.20; 95% CI 0.08–0.53) (Fig. 1C, D). Sensitivity analyses, censoring at 5, 6, 7, 8, 9 and 10 years, showed consistent results, with all HR estimations remaining consistently below 0.50 (Table 2). Additionally, there was no evidence of non-proportional hazards (p-value > 0.10 based on Schoenfeld residuals). In the multivariable model including age, tumor stage, nodal status, hormone receptor status and type of treatment, HER2DX risk-groups remained significantly associated with OS (HR = 0.30; 95% CI 0.11–0.84).

### HER2DX risk score in node-negative disease treated with trastuzumab alone
Among this population, the HER2DX risk score classified 57 tumors (85.1%) as low-risk and 10 tumors (14.9%) as high-risk. The 5-year RFS was 88.6% (95% CI 80.5–97.7) in the low-risk group versus 68.6% (95% CI 44.5–100) in the high-risk group (HR = 0.44; 95% CI 0.14–1.38). The 5-year OS was 96.4% (95% CI 91.6–100) in the low-risk group versus 80.0% (58.7–100) in the high-risk group (HR = 0.28; 95% CI 0.08–0.93). At 10 years, the RFS was 76.2% versus 57.1%, and the OS was 81.1% versus 45.7%, for low- and high-risk groups, respectively. Sensitivity analyses—censoring at 5, 6, 7, 8, 9, and 10 years—confirmed the prognostic consistency of the HER2DX risk score. Across all timepoints, HR estimates favored the low-risk group, remaining consistently below 0.50 for both RFS and OS (Table 3).

### HER2DX risk score, age and survival outcomes
Age was significantly associated with RFS (HR = 1.21; 95% CI 1.07–1.36; p = 0.002) and OS (HR = 1.26; 95% CI 1.09–1.46, p = 0.002) (Supplementary Tables 1, S2). The age distribution did not differ between patients with HER2DX high- and low-risk scores (Table 1). The median age in patients classified as HER2DX high and low risk score was 74 and 73 years, respectively (p = 0.20). After adjusting by age, similar results between HER2DX risk and survival outcomes were observed (RFS; all patients: HR = 0.57; 0.27–1.21; node-negative: HR = 0.42; 0.17–1.03) and (OS; all patients: HR = 0.42; 0.18–0.99; node-negative: HR = 0.21; 0.08–0.56).

### Biological signatures, TILs and prognostic value
The three biological signatures that compose the HER2DX risk score, proliferation, IGG/immune, and luminal, were evaluated independently. A heatmap of percentile scores across all 154 tumors profiled in Trans-RESPECT (Fig. 2A) illustrated substantial biological heterogeneity, highlighting the presence of immune-high, luminal-high, and proliferation-low subgroups. The figure also displays the HER2DX risk classification (low vs. high) for each sample, showing its relationship with the underlying biology. Modest pairwise correlations were observed among the signatures, including a positive correlation

**Table 1 | Clinical and pathological characteristics of patients in the RESPECT and trans-RESPECT cohorts**

|  | RESPECT | Trans-RESPECT | | | | |
|---|---|---|---|---|---|---|
|  | All patients | All patients | H arm | H + CT arm | HER2DX low-risk | HER2DX high-risk |
| N | 266 | 154 | 74 | 80 | 114 | 40 |
| Mean age | 73.9 | 73.7 | 73.6 | 73.9 | 73.6 | 74.3 |
| ER+ disease | 47.7% | 44.2% | 41.9% | 46.3% | 46.5% | 37.5% |
| pT stage |  |  |  |  |  |  |
| pT 1 | 48.9% | 51.3% | 52.7% | 50.0% | 61.4% | 22.5% |
| pT1b | 16.2% | 19.0% | 15.4% | 22.5% | 21.4% | 0.0% |
| pT1c | 83.8% | 81.0% | 84.6% | 77.5% | 78.6% | 100.0% |
| pT 2–3 | 51.1% | 48.7% | 47.3% | 50.0% | 38.6% | 77.5% |
| pT2 | 94.1% | 92.0% | 85.7% | 97.5% | 90.9% | 93.5% |
| pT3 | 5.9% | 8.0% | 14.3% | 2.5% | 9.1% | 6.5% |
| pN stage |  |  |  |  |  |  |
| pN 0 | 80.5% | 83.8% | 90.5% | 77.5% | 96.5% | 47.5% |
| pN 1–3 | 17.7% | 16.2% | 9.5% | 22.5% | 3.5% | 52.5% |
| Unknown | 1.9% | – | – | – | – | – |
| TILs |  |  |  |  |  |  |
| <20% | – | 63.4% | 63.0% | 63.8% | 59.3% | 75.0% |
| ≥20% | – | 36.6% | 37.0% | 36.2% | 40.7% | 25.0% |
| HER2DX pCR score |  |  |  |  |  |  |
| Med/Low | – | 48.7% | 44.6% | 52.5% | 53.5% | 55.0% |
| High | – | 51.3% | 55.4% | 47.5% | 46.5% | 45.0% |
| HER2DX ERBB2 score |  |  |  |  |  |  |
| Low | – | 20.1% | 20.3% | 20.0% | 12.5% | 22.8% |
| Med | – | 20.8% | 23.0% | 18.8% | 15.0% | 22.8% |
| High | – | 59.1% | 56.7% | 61.2% | 72.5% | 54.4% |

Baseline characteristics are shown for all patients enrolled in the RESPECT trial (n = 266) and for the subset included in the present biomarker analysis (Trans-RESPECT, n = 154), as well as stratified by treatment arm (trastuzumab alone [H] vs trastuzumab plus chemotherapy [H + CT]) and by HER2DX risk score group (low-risk vs high-risk).

*ER* estrogen receptor, *pT* pathological tumor size, *pN* pathological nodal status, *TILs* tumor-infiltrating lymphocytes, *NA* not available.

between proliferation and immune signatures ($r = 0.19$) and a weak inverse correlation between immune and luminal signatures ($r = -0.11$), supporting their biological complementarity.

To better understand the individual contribution of each biological component of the HER2DX risk score, univariable Cox models were used to examine their association with long-term outcomes. Each biological signature demonstrated hazard ratio estimates in the expected direction: high immune and luminal signatures were associated with lower risk of recurrence and death, whereas a high proliferation signature was associated with increased risk (Fig. 2B). These patterns were consistent across both the overall cohort and the node-negative subgroup and remained stable across multiple censoring timepoints (from 5 to 10 years and uncensored).

To explore the additive value of the biological signatures, a composite score ranging from 0 to 3 was created by assigning one point for each favorable profile (immune-high, luminal-high, proliferation-low). Increasing composite scores were associated with improved long-term outcomes, demonstrating a gradient of prognosis when combining these biological signals. Eight-year RFS and OS rates were progressively better with increasing composite scores, as shown in Kaplan–Meier analyses (Fig. 2C, D).

Finally, TILs were assessed as a conventional pathology variable, given their established prognostic relevance in early-stage HER2-

positive breast cancer. The median stromal TIL percentage was 12% (range: 0–80%), with 56 tumors (36.4%) exhibiting ≥20% TILs and 97 tumors (63.6%) showing <20%. In univariable Cox analyses, TILs analyzed as a continuous variable or dichotomized at the 20% threshold were not significantly associated with RFS or OS (Supplementary Tables 1–S4), although hazard ratio estimates were in the favorable direction (Fig. 2B). TILs were moderately correlated with the HER2DX immune signature (Pearson r = 0.66; Fig. 2E and Supplementary Fig. 4). In terms of HER2DX risk groups, tumors classified as low-risk were more likely to have high TIL levels: 40.7% of HER2DX low-risk tumors had ≥20% TILs, compared to 25.0% of HER2DX high-risk tumors (Table 1).

**HER2DX risk score, age, hormone receptor status and treatment interaction**

We evaluated whether the effect of chemotherapy varied according to HER2DX risk group, to assess potential treatment interaction. Among patients classified as HER2DX low-risk, the RFS HR for chemotherapy and trastuzumab versus trastuzumab alone was 0.49 (95% CI 0.18–1.30) and the OS HR was 0.47 (95% CI 0.14–1.55) for OS. The corresponding 10-year RFS and OS rates were 85.4% vs 76.7% and 90.2% vs 87.2%, respectively, for chemotherapy and trastuzumab versus trastuzumab alone. Among patients classified as HER2DX high-risk, the RFS HR was 1.45 (95% CI 0.43–4.83), with 10-year RFS rates of 64.6% vs 73.4%, and the OS HR was 1.14 (95% CI 0.32–4.06), with 10-year OS rates of 70.2% vs 68.7%, respectively. Overall, no statistically significant interaction was observed between HER2DX risk group and treatment arm. The interaction p-values for RFS and OS were 0.187 and 0.392, respectively. Similarly, age (<74 vs ≥74 years) was also not significantly associated with differential benefit from chemotherapy, with interaction p-values of 0.296 for RFS and 0.449 for OS. Finally, we also evaluated the relationship between hormone receptor status and treatment benefit; no significant interaction was observed for either RFS or OS (*p* for interaction = 0.770 and 0.901, respectively).

**HER2DX pCR score, prognosis and treatment interaction**

Among the 154 patients, 79 (51.3%) were classified as pCR-high, 33 (21.4%) as pCR-medium, and 42 (27.3%) as pCR-low based on the HER2DX pCR likelihood score. When evaluated as a prognostic marker, the pCR score was not significantly associated with either RFS or OS (Supplementary Figs. 5, 6).

When assessing treatment outcomes within these groups, a non-significant trend toward improved RFS was observed with chemotherapy in the pCR-high group (HR = 0.44, 95% CI 0.14–1.44), with 10-year RFS rates of 88.9% in the chemotherapy and trastuzumab arm and 61.4% in the trastuzumab-alone arm (Fig. 3A). Similarly, numerically better OS outcomes were noted in this group (HR = 0.23, 95% CI 0.05–1.08), with 10-year OS rates of 94.4% and 66.4%, respectively (Fig. 3B). In contrast, no evidence of benefit from chemotherapy was seen in patients classified as pCR-medium/low for either RFS (HR = 1.18, 95% CI 0.45–3.10; 10-year RFS: 70.6% with chemotherapy and trastuzumab vs 76.2% with trastuzumab alone) or OS (HR = 1.68, 95% CI 0.50–5.60; 10-year OS: 76.2% vs 88.6%) (Fig. 3C, D). The interaction p-values for RFS and OS were 0.200 and 0.045, respectively (Supplementary Tables 5, 6). Separate analysis by HER2DX pCR-low and medium group are provided in Supplementary Tables 7. These exploratory findings should be interpreted cautiously given the sample size and borderline significance.

## Discussion

This study shows that the HER2DX genomic assay provides long-term prognostic information in older patients with HER2+ early breast cancer treated with adjuvant trastuzumab with or without chemotherapy. In this population, standard clinicopathologic features, including tumor size, nodal status, hormone receptor expression, and

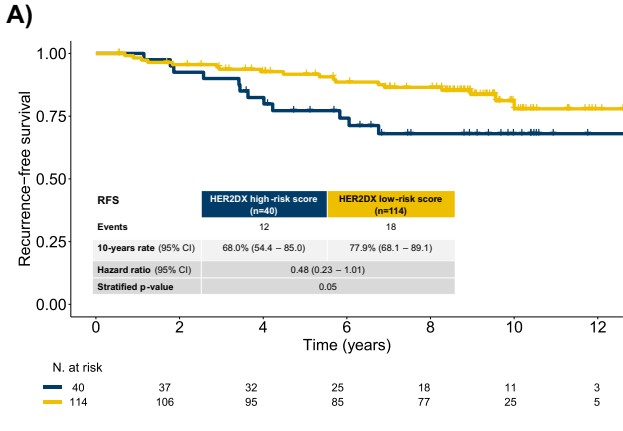

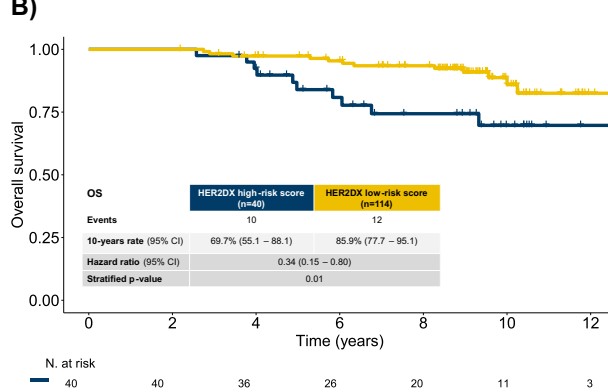

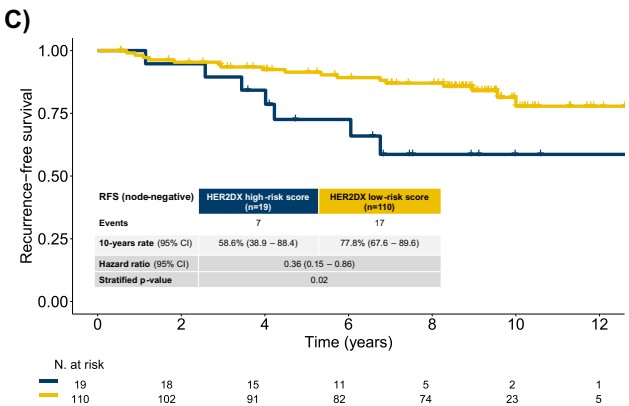

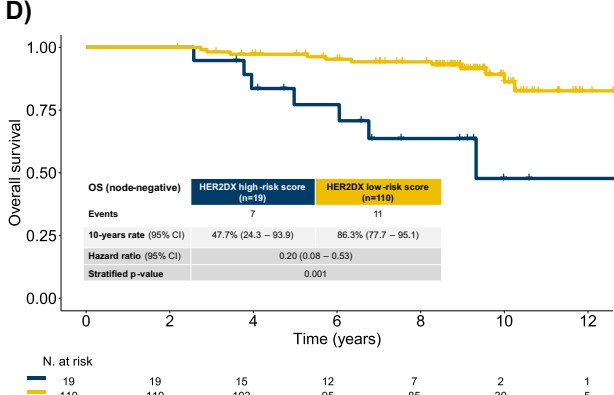

**Fig. 1 | Prognostic value of the HER2DX risk score for long-term outcomes.**
**A** Kaplan–Meier curve for RFS according to HER2DX risk group (low vs high) in all patients. **B** Kaplan–Meier curve for OS according to HER2DX risk group in all patients. **C** RFS by HER2DX risk group in the node-negative subgroup. **D** OS by HER2DX risk group in the node-negative subgroup. Hazard ratios (HRs), 95% confidence intervals (CIs), and p-values were calculated using Cox proportional hazards models. The HER2DX risk score was applied using prespecified cutoffs.

TILs, were not significantly associated with RFS or OS, highlighting the added value of genomic stratification by HER2DX.

While HER2DX has been previously validated in diverse early-stage HER2+ breast cancer populations, this study aimed to evaluate its prognostic and predictive performance in a randomized trial focused exclusively on older patients aged 70–80 years. Older adults remain underrepresented in clinical trials[3,6,28,29] and are frequently excluded from studies that inform biomarker development and treatment de-escalation strategies[30]. This gap has been highlighted by both the Breast International Group and expert guidelines from SIOG and EUSOMA, which underscore the lack of evidence supporting the use of multi-gene assays in older patients for either prognosis or prediction[30,31], as well as by the U.S. Food and Drug Administration, which has issued guidance specifically addressing the inclusion of older adults in cancer clinical trials[32]. Notably, even in the landmark EBCTCG meta-analysis of chemotherapy versus no chemotherapy, fewer than 5% of women were aged ≥70 years, underscoring the limited evidence base available to guide treatment decisions in this age group[33]. The recently published ASTER 70s trial, the first phase 3 randomized trial dedicated to women aged ≥70 years with breast cancer, showed that in ER-positive, HER2-negative disease with high genomic risk by the Genomic Grade Index, the addition of adjuvant chemotherapy to hormonotherapy did not confer a statistically significant survival benefit but was associated with substantially greater toxicity[34]. While ASTER 70s represents randomized evidence in older women with ER-positive, HER2-negative breast cancer, the RESPECT trial provides the randomized evidence in HER2+ disease, offering a unique opportunity to assess the role of HER2DX in guiding chemotherapy use

within this underrepresented population. These data therefore offer new insights into the potential role of HER2DX in informing chemotherapy use among older patients.

HER2DX identified a subgroup of patients with excellent long-term outcomes, particularly those with node-negative disease. These findings were consistent across censoring timepoints from 5 to 10 years, confirming the stability and clinical relevance of the assay's prognostic performance. The risk score's ability to capture underlying tumor biology was further supported by the associations of its three core gene expression signatures, immune/IGG, luminal, and proliferation, with survival outcomes.

In this study, hormone receptor status was not significantly associated with long-term outcomes, and no interaction was observed between hormone receptor status and treatment. Furthermore, HER2DX risk scores and underlying biological signatures were distributed across both hormone receptor-positive and -negative tumors, indicating that the assay captures biological information that is not merely a reflection of hormone receptor expression. These results align with (1) prior reports showing inconsistent associations between hormone receptor status and prognosis across studies[17,35,36], and (2) evidence that genomic stratification by HER2DX provides complementary prognostic information to standard clinicopathologic variables, including hormone receptor status, in early-stage HER2+ breast cancer[15,16,20,22,36].

The HER2DX pCR likelihood score is a genomic predictor developed to estimate the probability of achieving a pCR following neoadjuvant trastuzumab-based chemotherapy in early-stage HER2-positive breast cancer[15]. It reflects tumor-intrinsic features, such as proliferation,

**Table 2 | Sensitivity analyses of the HER2DX risk score for RFS and OS in the overall cohort and in the node-negative subgroup**

| Outcome | Group | Time (yrs) | Low-risk estimate (95% CI) | High-risk estimate (95% CI) | HR (95% CI) |
|---|---|---|---|---|---|
| RFS | All patients | 5 | 91.7 (86.7–97.1) | 77.2 (65.1–91.5) | 0.35 (0.1–0.9) |
| | | 6 | 88.6 (82.7–94.9) | 74.2 (61.6–89.4) | 0.41 (0.2–0.9) |
| | | 7 | 86.5 (80.1–93.4) | 68.0 (54.4–85.0) | 0.38 (0.2–0.8) |
| | | 8 | 86.5 (80.1–93.4) | 68.0 (54.4–85.0) | 0.38 (0.2–0.8) |
| | | 9 | 83.7 (76.6–91.5) | 68.0 (54.4–85.0) | 0.43 (0.2–0.9) |
| | | 10 | 81.2 (72.9–90.4) | 68.0 (54.4–85.0) | 0.46 (0.2–1.0) |
| | Node-negative | 5 | 91.4 (86.2–96.9) | 72.6 (54.6–96.4) | 0.30 (0.1–0.9) |
| | | 6 | 89.2 (83.4–95.2) | 72.6 (54.6–96.4) | 0.36 (0.1–1.0) |
| | | 7 | 87.0 (80.7–93.9) | 58.6 (38.9–88.4) | 0.28 (0.1–0.7) |
| | | 8 | 87.0 (80.7–93.9) | 58.6 (38.9–88.4) | 0.28 (0.1–0.7) |
| | | 9 | 84.1 (76.8–92.0) | 58.6 (38.9–88.4) | 0.31 (0.1–0.8) |
| | | 10 | 81.4 (72.8–90.9) | 58.6 (38.9–88.4) | 0.32 (0.1–0.8) |
| OS | All patients | 5 | 97.3 (94.4–100) | 83.9 (72.9–96.6) | 0.17 (0.0–0.7) |
| | | 6 | 95.4 (91.6–99.4) | 80.8 (68.9–94.8) | 0.23 (0.1–0.7) |
| | | 7 | 93.5 (88.9–98.3) | 74.3 (61.1–90.5) | 0.23 (0.1–0.6) |
| | | 8 | 93.5 (88.9–98.3) | 74.3 (61.1–90.5) | 0.23 (0.1–0.6) |
| | | 9 | 90.9 (85.4–96.9) | 74.3 (61.1–90.5) | 0.29 (0.1–0.7) |
| | | 10 | 88.7 (81.5–96.1) | 69.7 (55.2–88.1) | 0.30 (0.1–0.7) |
| | Node-negative | 5 | 97.2 (93.4–100) | 72.2 (59.7–99.8) | 0.12 (0.0–0.5) |
| | | 6 | 95.2 (91.3–99.6) | 70.8 (57.5–99.9) | 0.19 (0.0–0.9) |
| | | 7 | 94.2 (89.8–98.8) | 63.7 (43.9–92.3) | 0.14 (0.0–0.4) |
| | | 8 | 94.2 (89.8–98.8) | 63.7 (43.9–92.3) | 0.14 (0.0–0.4) |
| | | 9 | 91.6 (86.0–97.5) | 63.7 (43.9–92.3) | 0.18 (0.1–0.5) |
| | | 10 | 89.2 (82.3–96.7) | 47.7 (24.3–93.9) | 0.16 (0.1–0.4) |

These analyses are exploratory and were conducted to descriptively assess the consistency of the prognostic effect of the HER2DX risk score over time. They are not intended for formal statistical inference. Proportional hazards assumptions were verified in the primary model. Estimates were also derived for the node-negative subgroup.

**Table 3 | Sensitivity analyses of the HER2DX risk score for RFS and OS in patients with node-negative disease treated with trastuzumab alone**

| Outcome | Time (yrs) | Low-risk estimate (95% CI) | High-risk estimate (95% CI) | HR (95% CI) |
|---|---|---|---|---|
| RFS | 5 | 88.7 (80.5–97.7) | 68.6 (44.5–100.0) | 0.33 (0.1–1.3) |
| | 6 | 86.5 (77.6–96.4) | 68.6 (44.5–100.0) | 0.36 (0.1–1.5) |
| | 7 | 82.2 (72.2–93.5) | 57.1 (32.6–100.0) | 0.35 (0.1–1.2) |
| | 8 | 82.2 (72.2–93.5) | 57.1 (32.6–100.0) | 0.35 (0.1–1.2) |
| | 9 | 76.2 (64.4–90.2) | 57.1 (32.6–100.0) | 0.42 (0.1–1.3) |
| | 10 | 76.2 (64.4–90.2) | 57.1 (32.6–100.0) | 0.42 (0.1–1.3) |
| OS | 5 | 96.4 (91.6–100.0) | 80.0 (58.7–100.0) | 0.28 (0.1–0.9) |
| | 6 | 94.4 (88.5–100.0) | 80.0 (58.7–100.0) | 0.25 (0.0–1.5) |
| | 7 | 92.5 (85.6–99.9) | 68.6 (44.5–100.0) | 0.21 (0.1–0.9) |
| | 8 | 92.5 (85.6–99.9) | 68.6 (44.5–100.0) | 0.21 (0.1–0.9) |
| | 9 | 86.9 (77.3–97.7) | 68.6 (44.5–100.0) | 0.31 (0.1–1.2) |
| | 10 | 86.9 (77.3–97.7) | 45.7 (18.4–100.0) | 0.23 (0.1–0.8) |

At each timepoint, survival estimates and hazard ratios (HRs) were derived from models in which patients were censored at the specified year.

luminal, immune infiltration, and HER2 signaling, that are associated with chemosensitivity and response to HER2-targeted agents[2]. In this study, the pCR score identified a subgroup of patients (pCR-high) in which a numerical benefit from chemotherapy was observed. Specifically, patients classified as pCR-high appeared to derive greater benefit from the addition of chemotherapy to trastuzumab, particularly in terms of OS, whereas no such benefit was seen in the pCR-medium/low group.

These findings are biologically plausible, as tumors with a high HER2DX pCR score often display aggressive but chemo-sensitive features, including high proliferation, strong HER2 signaling, and robust immune infiltration[2,15,21]. Supporting this, a recent patient-level meta-analysis including 765 patients reported pCR rates of 80–90% in pCR-high tumors treated with a single taxane plus dual HER2 blockade[20]. In contrast, in the PHERGain trial, the same pCR-high tumors achieved only 44.6% pCR when treated with trastuzumab and pertuzumab alone without chemotherapy, despite PET-based patient selection and the use of endocrine therapy in hormone receptor–positive disease[12]. This suggests that the full therapeutic effect in these tumors relies on the combination of chemotherapy and HER2-targeted therapy. Although the interaction p-value for OS in RESPECT was nominally significant, these analyses were exploratory, based on small subgroups, and should be interpreted with caution.

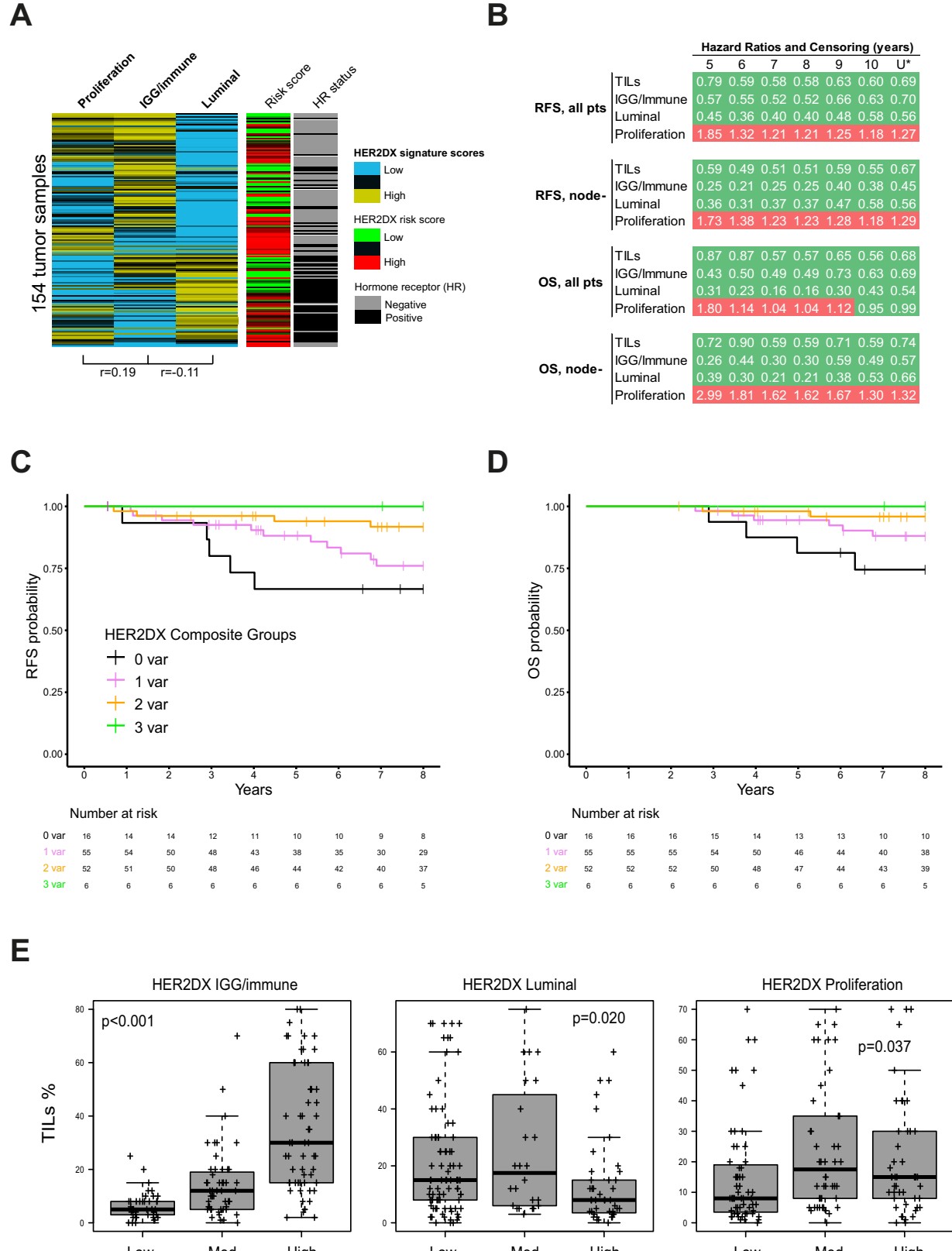

Interestingly, in Trans-RESPECT, 51.3% of tumors were classified as pCR-high—a proportion notably higher than the ~33% typically observed in neoadjuvant cohorts. A similar pattern was seen in the ATEMPT trial, where 78.6% of patients were pCR-high[16], suggesting that this distribution may be characteristic of early-stage, clinically low-risk HER2+ tumors treated in adjuvant de-escalation settings.

These findings raise the hypothesis that pCR-high biology is enriched in small, low-burden HER2+ cancers, possibly reflecting strong HER2 signaling and immune infiltration even in early-stage disease.

Notably, the pCR and risk scores were uncorrelated in this cohort (r = −0.009), confirming that HER2DX's prognostic and predictive components capture distinct biological dimensions[15]. Importantly,

**Fig. 2 | Prognostic value of individual HER2DX biological signatures and composite score. A** Heatmap of percentile scores for immune, luminal, and proliferation signatures across 154 tumors profiled in Trans-RESPECT, illustrating biological heterogeneity. Correlation coefficients (r) between two signatures are shown below the heatmap. In addition, HER2DX risk scores are shown for each patient. **B** Table showing hazard ratios (HRs) for RFS and OS across the three HER2DX biological signatures, immune/IGG, luminal, and proliferation, and TILs evaluated in the full cohort and in the node-negative subgroup. Sensitivity analyses are shown using different censoring timepoints (5–10 years and uncensored [U*]). Favorable HRs (below 1.0) are highlighted in green, while unfavorable HRs (above 1.0) are in red. **C** Kaplan–Meier curve showing 10-year RFS by composite score (0 to 3 favorable biological signatures). **D** Kaplan–Meier curve showing 10-year OS by composite score. The composite score assigns one point for each favorable feature: immune-high, luminal-high, and proliferation-low. **E** Boxplots showing the association between TIL levels and each HER2DX biological signature (IGG/immune, luminal, proliferation). *P* values are derived from Kruskal–Wallis tests.

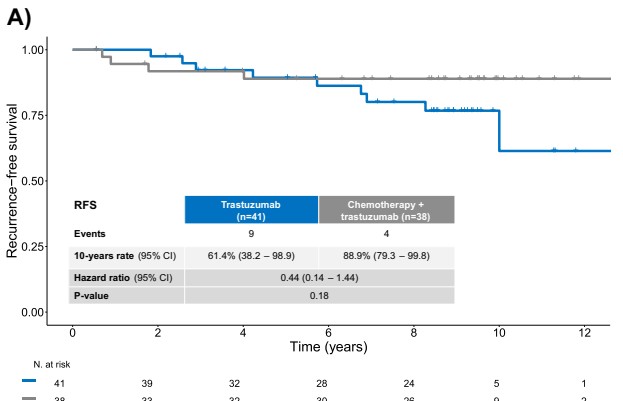

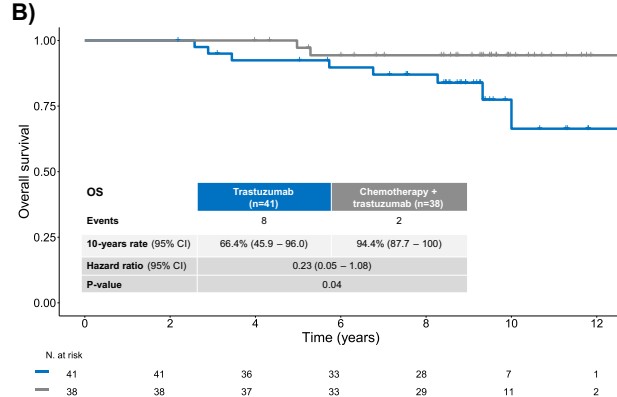

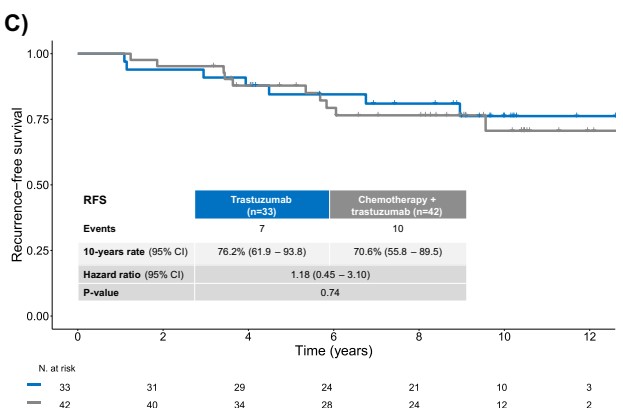

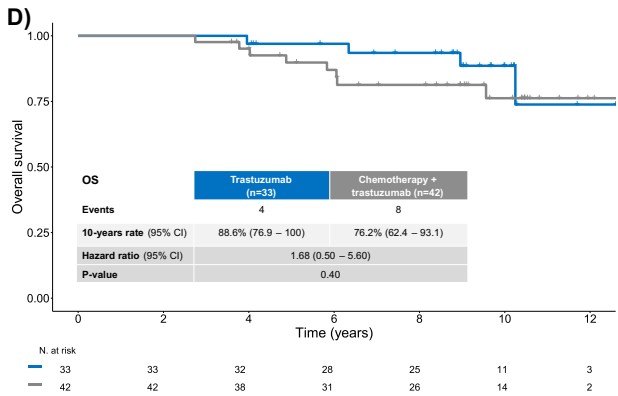

**Fig. 3 | Chemotherapy benefit according to HER2DX pCR likelihood score. A** Kaplan–Meier curves for RFS by treatment arm (trastuzumab alone vs trastuzumab + chemotherapy) in HER2DX pCR-high tumors. **B** Kaplan–Meier curves for OS by treatment arm in HER2DX pCR-high tumors. **C** RFS by treatment arm in HER2DX pCR-medium/low tumors. **D** OS by treatment arm in HER2DX pCR-medium/low tumors. Hazard ratios, 95% CIs, and *p*-values were derived from univariable Cox models. Interaction *p*-values reflect formal tests for treatment–biomarker interaction.

34% of patients in this cohort were classified as both HER2DX low-risk and pCR-medium/low. This overlap between the prognostic and predictive components of HER2DX suggests that combining these two scores may offer complementary insights when tailoring treatment strategies. In particular, this dual-classification approach could help identify older individuals with HER2-positive early breast cancer who may be appropriate candidates for de-escalated therapies, including the omission of cytotoxic chemotherapy. However, these results should be interpreted with caution given the limited sample size and exploratory nature of the subgroup analyses.

Neoadjuvant systemic therapy is considered the current standard of care for patients with stage II–III HER2+ breast cancer, as it provides prognostic information through assessment of pCR and enables tailoring of adjuvant therapy[37]. In the RESPECT trial[9], ~43.0% of patients had stage I disease, for which neoadjuvant therapy is not standard, while 57.0% had stage II–III disease (41.5% stage IIA, 13.3% stage IIB, and 1.1% stage IIIA), which could have been candidates for neoadjuvant therapy if primary surgery had not been performed. However,

neoadjuvant therapy in this setting typically consists of multi-agent chemotherapy regimens such as AC-T or docetaxel–carboplatin[37], which have substantial toxicity, and, if pCR is not achieved, these patients are subsequently exposed to adjuvant T-DM1[37,38]. In contrast, in RESPECT, no patient received these regimens[9]. At the time the RESPECT trial was initiated, residual disease–guided treatment following neoadjuvant chemotherapy was not recommended. The trial also pre-dated the approval of neoadjuvant pertuzumab and the results of the APHINITY trial[39], which demonstrated an overall survival benefit with the addition of pertuzumab in node-positive disease[40]. Nonetheless, the proportion of patients with node-positive disease in RESPECT was low (16%)[9].

Beyond its utility in guiding adjuvant treatment decisions, the HER2DX risk score has also been proposed as a tool to support initial therapeutic planning, specifically, the choice between upfront surgery and neoadjuvant systemic therapy. Recent expert recommendations[18] suggest that patients with HER2DX low-risk tumors, particularly those also classified as pCR-medium/low, may be appropriate candidates for

primary surgery, thereby minimizing exposure to potentially unnecessary neoadjuvant treatment. Conversely, individuals with HER2DX high-risk or pCR-high tumors may derive greater benefit from neoadjuvant strategies[18]. Our findings are consistent with this proposed framework and may be particularly relevant for older patients, in whom treatment burden and tolerability are critical considerations. Importantly, HER2DX can be reliably performed using RNA extracted from standard FFPE core needle biopsies obtained at diagnosis[13,15,20], without the need for additional tissue, enabling its integration into pre-treatment decision-making[18]. However, we acknowledge that prospective data to guide surgical sequencing decisions remain limited and are unlikely to be generated systematically, underscoring the importance of individualized, multidisciplinary decision making.

The Trans-RESPECT validation of the HER2DX risk score adds to the growing body of evidence supporting this genomic test[10–13,15–21,24,36,41–43]. The score was initially developed using clinical and genomic data from 434 patients enrolled in the Short-HER phase III trial[15], which randomized individuals with stage I–III HER2+ early breast cancer to receive either 9 weeks or 1 year of adjuvant trastuzumab in combination with multi-agent chemotherapy. The HER2DX risk score integrates three gene expression signatures—immune/B-cell infiltration, tumor proliferation, and luminal differentiation—together with tumor size and nodal status, to generate a continuous score ranging from 1 to 99[15,18]. Prespecified cutoffs were established to define low- and high-risk groups, with the low-risk group designed to achieve >90% distant recurrence-free survival at 3, 5, and 7 years, independent of trastuzumab duration[15]. Since its development, the HER2DX risk score has been externally validated in both adjuvant and neoadjuvant settings[42]. It has shown consistent prognostic performance even in low-event cohorts, such as the APT[17] and ATEMPT[16] trials, and provides prognostic information that is independent of pCR status in the neoadjuvant setting[20]. Its overall performance has also been confirmed in a patient-level meta-analysis including 2518 patients with stage I–III HER2+ disease[36].

This study has several limitations. The analysis was retrospective and limited to a subset of RESPECT trial participants, and the sample size, although among the largest datasets with genomic profiling in older patients with HER2+ breast cancer, remains modest, particularly for subgroup and interaction analyses. Although HER2DX was applied using predefined thresholds and genomic testing was conducted blinded to clinical data, treatment assignment was not guided by the assay. In addition, the chemotherapy arm of the original trial included a variety of regimens, most commonly weekly paclitaxel monotherapy (35.1%), anthracycline-based combinations (22.9%), CMF (cyclophosphamide, methotrexate, and fluorouracil; 19.8%), docetaxel monotherapy (14.5%), and TC (docetaxel and cyclophosphamide; 3.1%)[9]. While all these regimens are considered standard, many patients in current clinical practice—especially those who are older and/or have stage I (clinically low-risk) disease—would likely receive single-agent paclitaxel, as used in the APT trial[17]. Therefore, our findings cannot be used to infer the benefit of any specific chemotherapy backbone. Finally, our findings are specific to the HER2DX assay and cannot be generalized to other commercially available genomic tests such as Oncotype DX, MammaPrint, EndoPredict, or Prosigna, which differ in development context, gene content, and biological focus[44]. In addition, HER2DX was designed specifically for HER2+ early breast cancer and uniquely incorporates an immune-related gene signature, which is particularly relevant in predicting prognosis and response to HER2-targeted therapies[15,17,45]. Future comparative studies between HER2DX and other genomic assays in HER2+ populations may help clarify differences in clinical utility and biological coverage across platforms. Lastly, the RESPECT trial population was highly specific, with a median patient age of 74 years, 9 years of follow-up, and a remarkably low incidence of non-cancer-related deaths. This likely reflects two factors: (i) the high life expectancy among women in Japan (87.1 years), and (ii)

the selection of relatively fit older adults eligible for randomization, who probably had fewer comorbidities than the general older population. These features may limit the generalizability of our findings to settings with shorter life expectancy or higher competing risks of non-cancer mortality. However, life expectancy continues to increase in many developed countries[46], and cancer incidence itself rises steeply with age: currently, patients aged ≥65 years account for approximately half of all cancer diagnoses and those ≥80 years for ~13%, proportions expected to further increase in the coming decades[47]. Breast cancer is the most commonly diagnosed cancer in older women, with a peak incidence near age 65 in many countries[47]. A substantial proportion of these individuals remain in good functional status, underscoring the clinical relevance of our findings and the need for tools such as HER2DX to support tailored treatment decisions in this growing population.

In conclusion, HER2DX provides long-term prognostic information in older patients with HER2+ early breast cancer. When combined with the HER2DX pCR score, it may help identify individuals most likely, and least likely, to benefit from chemotherapy, supporting a more personalized approach to treatment in this understudied population.

### Reporting summary
Further information on research design is available in the Nature Portfolio Reporting Summary linked to this article.

### Data availability
The protocol of the N-SAS BC 07 (RESPECT) study is available as a Supplementary file (Supplementary Note 1). The clinical and genomic data generated and analyzed in this study are not publicly available due to intellectual property and licensing restrictions related to the HER2DX assay. However, data can be made available to academic researchers upon request, subject to a data transfer agreement and approval by the data custodians. Requests should be directed to alprat@clinic.cat. Source data are provided with this paper.

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

## Acknowledgements

We sincerely thank the patients and their families, as well as the investigators and research teams whose contributions and participation made this study possible. This study was funded by CSPOR, Chugai Pharmaceutical Co. Ltd and Reveal Genomics. This study was supported by the Comprehensive Support Project for Oncology Research (CSPOR) and Reveal Genomics. Data from this study will be presented in part at the 2025 American Society of Clinical Oncology (ASCO) Annual Meeting.

## Competing interests

G.V. has received speaker fees from MSD, Pfizer, GSK and Pierre Fabre, has held an advisory role with AstraZeneca, and received consultant fees from Reveal Genomics. AV reports personal fees from Bayer, Bristol Myers Squibb, Guardant Health, Merck, Novartis, Roche, and Incyte, outside the submitted work. F.B.M. has a patent PCT/EP2021/070788 (In vitro method for the prognosis of patients suffering from her2-positive breast cancer) licensed to Reveal Genomics. J.S.P., C.M.P. and A.P. are equity stockholders and consultants of Reveal Genomics. L.P. has a patent WO2018/103834 (Her2 as a predictor of response to dual HER2 blockade in the absence of cytotoxic therapy) licensed to Reveal Genomics. A.P. reports grants from Reveal Genomics during the conduct of the study, personal fees from Roche, and grants and personal fees from AstraZeneca, Daiichi Sankyo, and Novartis, outside the submitted work; in addition, A.P. has patent WO2018/103834 (Her2 as a predictor of response to dual HER2 blockade in the absence of cytotoxic therapy), PCT/EP2021/070788 (In vitro method for the prognosis of patients suffering from her2-positive breast cancer) licensed to Reveal Genomics. All other authors declare no competing interests.

## Additional information

[1]Department of Advanced Clinical Research and Development, Nagoya City University Graduate School of Medical Sciences, Nagoya, Japan. [2]Department of Breast Surgery, Nagoya City University Graduate School of Medical Sciences, Nagoya, Japan. [3]Department of Breast Oncology, Nagoya Medical Center, Nagoya, Japan. [4]Biostatistics Section, Department of Data Science, Center of Clinical Sciences, National Center for Global Health and Medicine, Shinjuku, Japan. [5]Department of Breast Surgery, Shizuoka General Hospital, Shizuoka, Japan. [6]Department of Breast Medical Oncology, The Cancer Institute Hospital of JFCR, Koto, Japan. [7]Department of Breast and Thyroid Surgical Oncology, Sagara Hospital, Kagoshima, Japan. [8]Department of Breast Oncology, Aichi Cancer Center Hospital, Nagoya, Japan. [9]Department of Breast Oncology, Jichi Medical University, Shimotsuke, Japan. [10]Department of Breast and Thyroid Surgery, Kawasaki Medical School, Kurashiki, Japan. [11]Reveal Genomics, Barcelona, Spain. [12]Department of Genetics, Lineberger Comprehensive Cancer Center, University of North Carolina, Chapel Hill, NC, USA. [13]Statistics Unit, Vall d'Hebron Institute of Oncology, Barcelona, Spain. [14]Translational Genomics and Targeted Therapies in Solid Tumors, August Pi i Sunyer Biomedical Research Institute (IDIBAPS), Barcelona, Spain. [15]Department of Medical Oncology, Clínic Barcelona Comprehensive Cancer Center, Barcelona, Spain. [16]Breast Cancer Unit, IOB-QuirónSalud, Barcelona, Spain. [17]Department of Medicine, University of Barcelona, Barcelona, Spain. [18]These authors jointly supervised this work: Aleix Prat, Hiroji Iwata. ✉e-mail: alprat@clinic.cat

