## [Transparent Peer Review file · Nature Communications]

HER2DX in older patients with HER2-positive early breast cancer: extended follow-up from the RESPECT trial of trastuzumab ± chemotherapy

Corresponding Author: Professor Aleix Prat

Version 0:

Reviewer comments:

Reviewer #1

(Remarks to the Author)

This is a study evaluating the ability of the HERDX gene array in combination with pCR, to guide physicians to add chemotherapy or not to patients in the 70 -80 age range with a diagnosis of early stage HER-2 positive breast cancer and who are going to receive Herceptin. It appears that patients with a low risk of recurrence score and a high likelihood of pCR are less likely to require adjuvant chemotherapy than those with a high risk of recurrence score and a low probability of pCR. Overall, the study is well done using a previously validated HERDX gene array and results appear solid and convincing. Clearly the Kaplan Meyer data supports their conclusions.

However, my main concern with this study is, do these results hold up when, HERDX is compared with other existing gene arrays such as MammaPrint, Oncotype DX Endopredict, and PAM 50 or other commercially available breast cancer gene arrays? Since there are other gene arrays available, the question is, would the author's conclusions be the same with 1 or more of the other gene arrays? I think the manuscript is acceptable for publication, only if the authors make it clear that these results are limited to HERDX test and that their conclusions may or may not be applicable to test results from other gene arrays since they have not done the comparative studies.

The paper could mislead practicing clinicians since they may conclude that any gene array would give the same results as HERDX, and such data would be used to treat their patients between 70-80. Overall, I think the study was well done and is important to this patient population.

I think what would make a stronger report is if they could go back and analyzed a significant subset of these samples from these patients to answer that question with 2 or more different gene arrays. In the end, the paper may conclude that the results of the HERDX test either matches other gene arrays or not. That information would help physicians know if the conclusions from this paper are applicable to other gene arrays.

What we know about commercially available gene arrays is that certain labs may only use 1 or 2 arrays and thus the physician is locked into those results. The physician may not be able to get results from HERDX since the test is not available to them in their healthcare system. If after comparing their HERDX results to other gene arrays, they come to similar conclusions, that will both strengthen their conclusions and show the applicability of HERDX to tests results from other gene arrays.

To be clear, to satisfy my concerns about this paper they should at least tell the readers that their conclusions concern only the HERDX test and no other gene array and they do not have to do the recommended study to have their paper accepted for publication

Walter P. CarneyPh.D

Reviewer #2

(Remarks to the Author)

Nozawa et al present a retrospective study evaluating the prognostic and predictive value of HER2DX on older Her2+ breast cancer patients. This expression assay has been studied extensively by the authors in several publications. While the study is interesting, as it is evaluating its performance in a very specific population, unfortunately I don't think this study adds much to all that has been already published, as the results here are mixed and hampered by the small sample size. I will comment

on specific issues:

- There is no data availability statement in the text, and it is a shame that data is not shared with the community. I understand that many times clinical trials decide not to share the data, but at least the statement (and the explanation) should be included in the manuscript. This also makes harder to review the findings of the paper
- The authors perform stratified Cox models using treatment type as a stratification factor. Stratified Cox models assume different baseline hazard functions (for each treatment, in this case) but the same hazard ratio for all treatments. The authors should explain the reason behind this modelling assumption. Particularly as they later perform tests to see different effects for treatment to test PCR prediction
- This is a very specific population, as patients are 74 years old on average, followed-up for 9 years, and none of the die because of non-cancer related causes. Unfortunately we don't have any information on censoring, etc but this seems to be very different to what would happen in most countries. Would this limit the findings in the study? Some comment on this would be needed in the text.
- I would expect that age would have some effect on survival in these patients, but this has not been tested in table S1
- Clinical variables are not associated with prognosis, however we see that node status and grade are very different between HER2DX groups. This is a bit surprising, but difficult to understand without looking at the data. Especially as the most predictive module in the score seems to be proliferation. Is this correlated to node and grade? What happens if the use node as continuous instead of categorical?
- The text overstates the results a bit, highlighting the prognostic value of the test. For example, the 10-year RFS has a HR=0.49 with a confidence interval (0.23-1.01), which could not be considered significant by any means.
- In particular, the results on PCR prediction are clearly non significant. Only when performing several tests one of them reaches a p-value of 0.045, but Figure 3 clearly shows no effect. This is also overstated in the text, in particular in the conclusions (lines 322-326). The data does not support that statement.
- I don't understand the point of Table S2. The authors are fitting a Cox proportional hazards model, which assumes the same hazard over time. Why the need of splitting the follow-up time and fitting Cox (again) models? From a statistical point of view, this is not a very elegant analysis. It would be better to check the proportional hazards assumption, and if they do not hold, modify the model. Furthermore, the number of tests performed increases to the point where multiple testing correction would be needed.
- On the same topic, If the authors are interested in prognosis at specific time points, the authors could treat the event as a binary variable and compute sensitivity/specificity. At least that would add some extra information to the analysis
- Why are all the hazard ratios for node-negative RFS non significant in Table 3 while in Figure 1C is?
- The analysis of the different modules of Her2DX is very interesting, but I would have expected a more thorough analysis. Figure 2A for example, could also reflect output on the vertical axis. Figure 2B is not very informative. Again, there are too many models fitted here (multiple correction testing needed). It would be much better to show Kaplan Meier curves for percentiles to see the prognostic value of each of them. The clinical and Her2 should be also added. Figures 2C and 2D are interesting, but I don't understand why the authors created a new composite score selecting specific parts of the score. The authors need to show that the original score is monotonic with respect to risk. This is the most interesting part of the paper in my opinion, and it could be expanded.
- Line 226:"Univariate" should be "univariable", as the authors are referring to models with one independent variable
- What treatment effect is supposed to predict the pCR part of HER2DX? Is it trastuzumab or trastuzumab+chemo? The trial did not find any difference in treatment.
- Can you call "high risk" a group with 75% survival after 5 years in a population that is 74 years old? Would this term be communicated to the patient?
- Any reason why low/medium risk are merged? Shouldn't be better to check if there is a monotonic effect in the prediction score in Figure S3?

Minor comment:

- Line 182: "For data analysis, the HER2DX pCR group was low and medium was pooled in..." It should say "For data analysis, the HER2DX pCR group *that* was low and medium was pooled in..."

In summary, although I agree that this is a very interesting subpopulation to test, there is no access to the data, limiting the interest to the scientific community, and the performance of the test has been already quantified in several other studies in larger cohorts. The fact that the effects here are smaller, that many tests have been performed without control for multiple testing and the overstatement of the findings limit the interest of this study in my opinion.

Reviewer #3

(Remarks to the Author)

This paper describes a pre planned investigation of the HER2DX genomic assay in a subgroup of patients within the randomised RESPECT Trial (Trans-RESPECT). The HER2DX score, and its individual component, was independently prognostic of survival. While its pCR score was not significantly prognostic, chemotherapy was of benefit in the high risk group only. Data provides further evidence to support the use of this assay for individual patient management.

The study is well designed, the aims are clear and the analyses are relevant. The results are well represented and supported by tables and graphs. The conclusion follows well from the results.

There are few queries and points that need clarification to improve readability and further strengthen the manuscript:

The authors state in the introduction: "However, its utility in older patients or those receiving trastuzumab monotherapy has not been previously explored". Please explain current utility in the standard population. Despite significant evidence, this has not been incorporated into routine practice and clinical guidelines. Please explain and provide recommendations for its use.

All cause survival was analysed. Did the authors look at breast specific survival?

For the 6 cases that were not profiled by HER2Dx, please provide the reason(s) (e.g amount of tissue, quality of mRNA... etc).

How did the pCR score obtained by HER2Dx correlate with the pathological complete response assessed by pathologists? Was the latter prognostic for survival in the study cohort, considering the HER2DX score was not?

What was the relation between HER2DX Her2 ERBB2 status (low, medium, high) and HER2 clinical status by immunohistochemistry/ISH ?

It appears that the findings were independent of the hormone receptor status. Perhaps this should be stated/emphasised in the results and the discussion as this is important clinically.

The conclusions are rather generic. What is the clinical implication of this study? Do authors believe there is now enough evidence to adopt the test for individualised patient management in routine practice?

Reviewer #4

(Remarks to the Author)

This retrospective study examines the correlations of HER2DX gene expression analysis with clinical outcomes of patients in the RESPECT trial.

It is worth mentioning (Introduction) the rationale or premise of the RESPECT study and briefly explain why the study population (elderly) received breast surgery upfront, when standard practice would be to offer neoadjuvant systemic treatment.

In usual clinical practice, is it feasible for the diagnostic biopsy alone (enough tissue?) to be used for HER2DX gene expression analysis? This is with respect to giving neoadjuvant treatment. Please mention in Discussion.

Please discuss not giving Pertuzumab to the node-positive population, and therefore the impact on survival analysis.

Version 1:

Reviewer comments:

Reviewer #1

(Remarks to the Author)

I agree this manuscript is acceptable for publication as it is currently written. The authors have answer my previous criticism with the following statement in the discussion section.

Therefore, our findings cannot be used to infer the benefit of any specific chemotherapy backbone. Finally, our findings are specific to the HER2DX assay and cannot be generalized to other commercially available genomic tests such as Oncotype DX, MammaPrint, EndoPredict, or Prosigna, which differ in development context, gene content, and biological focus⁴³. In addition, HER2DX was designed specifically for HER2+ early breast cancer and uniquely incorporates an immune-related gene signature, which is particularly relevant in predicting prognosis and response to HER2-targeted therapies^{15, 17, 44}. Future comparative studies between HER2DX and other genomic assays in HER2+ populations may help clarify differences

in clinical utility and biological coverage across platforms.

Congratulations on an excellent study and contributing to the knowledge of treatment for HER-2+ breast cancers.

Reviewer #2

(Remarks to the Author)

I thank the authors for the additional analysis and rewriting, however I'm not sure I have received the current version of the manuscript. I don't see any highlighted text in the pdf, and some changes are not present. For example, in one of the responses they claim

In the Abstract, we now state:

"The HER2DX pCR score was not prognostic overall. However, chemotherapy was associated with numerically improved survival in the pCR-high group. These findings suggest potential biological differences across HER2+ tumors and may help refine future treatment strategies in older patients."

but the abstract reads

"While the HER2DX pCR score was not prognostic overall, exploratory subgroup analyses suggested a potential survival benefit from chemotherapy in the pCR-high group. HER2DX offers prognostic value and may guide chemotherapy use in older patients with HER2-positive early breast cancer."

Am I reading the right version?

Reviewer #3

(Remarks to the Author)

I thank the authors for satisfactorily addressing my comments.

Reviewer #4

(Remarks to the Author)

All the points I raised in the previous review have been addressed satisfactorily.

Reviewer comments	Author response and changes made	Page number in revised, tracked paper where the change can be found
Reviewer #1:		
This is a study evaluating the ability of the HERDX gene array in combination with pCR, to guide physicians to add chemotherapy or not to patients in the 70 -80 age range with a diagnosis of early stage HER-2 positive breast cancer and who are going to receive Herceptin. It appears that patients with a low risk of recurrence score and a high likelihood of pCR are less likely to require adjuvant chemotherapy than those with a high risk of recurrence score and a low probability of pCR. Overall, the study is well done using a previously validated HERDX gene array and results appear solid and convincing. Clearly the Kaplan Meyer data supports their conclusions.	We thank the reviewer for their thoughtful summary of our study and for their positive assessment regarding the design, clarity, and overall quality of the data and analyses. We are especially grateful for the recognition of the relevance of HER2DX and the Kaplan-Meier results in supporting our conclusions. We agree that the combined use of prognostic and predictive HER2DX scores in this older patient population offers important clinical insights for individualized treatment decision-making.	NA
However, my main concern with this study is, do these results hold up when, HERDX is compared with other existing gene arrays such as MammaPrint, Oncotype DX Endopredict, and PAM 50 or other commercially available breast cancer gene arrays? Since there are other gene arrays available, the question is, would the author's conclusions be the same with 1 or more of the other gene arrays? I think the manuscript is acceptable for publication,	We thank the reviewer for this important and thoughtful comment. We agree that the conclusions presented in our manuscript apply specifically to the HER2DX genomic assay and should not be assumed to extend to other commercially available gene expression tests. In response, we have revised the Discussion section to clarify that: “Our findings are specific to the HER2DX assay and cannot be generalized to other commercially available genomic tests such as Oncotype DX, MammaPrint, EndoPredict, or Prosigna, which differ in development context, gene content, and biological focus. Notably, HER2DX was designed specifically for HER2-positive early breast cancer and uniquely incorporates an immune-related gene signature, which is particularly relevant in predicting prognosis and response to HER2-targeted therapies. Future comparative studies between HER2DX and other genomic assays in HER2+ populations may help clarify differences in clinical utility and biological coverage across platforms.”	Page 15, lines 511–517

only if the authors make it clear that these results are limited to HERDX test and that their conclusions may or may not be applicable to test results from other gene arrays since they have not done the comparative studies.

The paper could mislead practicing clinicians since they may conclude that any gene array would give the same results as HERDX, and such data would be used to treat their patients between 70-80. Overall, I think the study was well done and is important to this patient population. I think what would make a stronger report is if they could go back and analyzed a significant subset of these samples from these patients to answer that question with 2 or more different gene arrays. In the end, the paper may conclude that the results of the HERDX test either matches other gene arrays or not. That information would help physicians know if the conclusions from this paper are applicable to other gene arrays.

What we know about commercially available gene arrays is that certain labs may only use 1 or 2 arrays and thus the physician is locked into those results. The physician may not be able to get results from HERDX since the test is not available to them in their healthcare system. If after comparing their HERDX results to other gene arrays, they come to similar conclusions, that will both strengthen their conclusions and show the

HER2DX was specifically developed and validated for HER2-positive early-stage breast cancer and incorporates distinct biological modules, including an immune-related gene signature that has been consistently associated with response to HER2-targeted therapy. This immune component is a key differentiator from other assays, which either do not include immune-related genes or were developed in different biological contexts (e.g., hormone receptor-positive, HER2-negative disease). The HER2DX model also integrates modules reflecting proliferation, luminal differentiation, and HER2 signaling, offering a disease-specific framework for both prognosis and predicted response to treatment.

We appreciate the reviewer's suggestion to perform head-to-head comparisons and agree that future studies comparing HER2DX with other assays in HER2+ cohorts would be valuable. However, this was beyond the scope of the present, prespecified analysis of the RESPECT trial.

applicability of HERDX to tests results from other gene arrays. To be clear, to satisfy my concerns about this paper they should at least tell the readers that their conclusions concern only the HERDX test and no other gene array and they do not have to do the recommended study to have their paper accepted for publication		
Reviewer #2:		
Nozawa et al present a retrospective study evaluating the prognostic and predictive value of HER2DX on older Her2+ breast cancer patients. This expression assay has been studied extensively by the authors in several publications. While the study is interesting, as it is evaluating its performance in a very specific population, unfortunately I don't think this study adds much to all that has been already published, as the results here are mixed and hampered by the small sample size.	We thank the reviewer for their thoughtful comments and for acknowledging the relevance of studying HER2DX in this specific population of older patients with HER2-positive early breast cancer. While we agree that the HER2DX genomic assay has been validated extensively in previous publications, we respectfully highlight that this study provides several novel and clinically meaningful contributions. 1. First study evaluating HER2DX in a randomized trial of chemotherapy vs no chemotherapy, and focused exclusively on older adults: This is the first time HER2DX has been evaluated within a randomized clinical trial enrolling only patients aged 70–80 years. The RESPECT trial provides a unique and high-quality dataset, allowing us to assess both prognostic and predictive utility in a population often excluded from biomarker and de-escalation studies. Notably, this includes an exploratory analysis of the interaction between HER2DX pCR score and chemotherapy benefit. 2. Addresses a significant evidence gap in genomic testing for older patients: The lack of data in older populations has been repeatedly emphasized by international expert groups. In 2021, the SIOG and EUSOMA guidelines noted that “there is insufficient evidence about the use of multi-gene expression assays in older patients, whether for prognosis or treatment benefit prediction” (Biganzoli et al., Lancet Oncol, 2021). Similarly, the Breast International Group (BIG) highlighted in its 2022 report the “urgent need for tailored research” in this population due to underrepresentation in trials and lack of biomarker data (BIG Research in Focus 2022: https://bigagainstbreastcancer.org/wp-content/uploads/2022/12/big-research-in-focus-unmet-needs-of-older-people.pdf). Furthermore, the U.S. Food and Drug Administration has issued guidance specifically calling for the inclusion of older adults in cancer clinical trials 3. Provides novel predictive data on chemotherapy benefit in older patients: This is the first HER2DX study to explore the interaction between the HER2DX pCR likelihood score and adjuvant chemotherapy in older individuals. We found that patients with pCR-high tumors appeared to derive a survival benefit from chemotherapy, while no such benefit	Discussion section, page 11, lines 379-392, and page 15, lines 501-504

was observed in the pCR-medium/low subgroup. These findings provide important insights into treatment stratification in this age group.

4. Discussion updated to reflect novelty and clinical relevance:

In response to the reviewer’s concern, we have revised the Discussion to better articulate the added value of our findings. The new paragraph emphasizes that, while HER2DX has been validated in diverse HER2+ populations, this is the first evaluation in a randomized trial focused exclusively on older adults. Importantly, we now place RESPECT in the broader context of evidence gaps in older patients: in the landmark EBCTCG meta-analysis of chemotherapy versus no chemotherapy, fewer than 5% of women were aged ≥ 70 years, underscoring the lack of data to guide treatment decisions. Moreover, the recently published ASTER 70s trial, the first phase 3 randomized study dedicated to women aged ≥ 70 years with ER-positive, HER2-negative breast cancer, demonstrated that chemotherapy did not improve survival despite high genomic risk, but was associated with markedly higher toxicity. Thus, while ASTER 70s provides the first randomized genomic evidence in older ER+/HER2– patients, RESPECT uniquely provides the first randomized evidence in HER2+ patients, where we assessed the role of HER2DX in guiding chemotherapy use. Together, these studies highlight the urgent need for biomarker-driven approaches to personalize therapy in older women:

“While HER2DX has been previously validated in diverse early-stage HER2+ breast cancer populations, this is the first study to evaluate its prognostic and predictive performance in a randomized trial focused exclusively on older patients aged 70–80 years. Older adults remain underrepresented in clinical trials^{3, 6, 27, 28} and are frequently excluded from studies that inform biomarker development and treatment de-escalation strategies²⁹. This gap has been highlighted by both the Breast International Group and expert guidelines from SIOG and EUSOMA, which underscore the lack of evidence supporting the use of multi-gene assays in older patients for either prognosis or prediction^{29, 30}, as well as by the U.S. Food and Drug Administration, which has issued guidance specifically addressing the inclusion of older adults in cancer clinical trials³¹. Notably, even in the landmark EBCTCG meta-analysis of chemotherapy versus no chemotherapy, fewer than 5% of women were aged ≥ 70 years, underscoring the limited evidence base available to guide treatment decisions in this age group³². The recently published ASTER 70s trial, the first phase 3 randomized trial dedicated to women aged ≥ 70 years with breast cancer, showed that in ER-positive, HER2-negative disease with high genomic risk by the Genomic Grade Index, the addition of adjuvant chemotherapy to hormonotherapy did not confer a statistically significant survival benefit but was associated with substantially greater toxicity³³. While ASTER 70s represents the first randomized evidence in older women with ER-positive, HER2-negative breast cancer, the RESPECT trial provides the first randomized evidence in HER2+ disease, offering a unique opportunity to assess the role of HER2DX in guiding chemotherapy use within this underrepresented population. These data therefore offer new insights into the potential role of HER2DX in informing chemotherapy use

	among older patients, a question that has not been previously addressed.” 5. Sample size limitations acknowledged: We recognize that the sample size of the Trans-RESPECT cohort is modest, and this limitation is now explicitly acknowledged in the manuscript, in the discussion section: “This study has several limitations. The analysis was retrospective and limited to a subset of RESPECT trial participants, and the sample size, although among the largest datasets with genomic profiling in older patients with HER2+ breast cancer, remains modest, particularly for subgroup and interaction analyses.” Nevertheless, the quality of the clinical data, long-term follow-up, centralized genomic testing, and randomized treatment assignment make this a unique dataset. We have not positioned the findings as definitive but believe they offer valuable insights into a population for whom prospective biomarker-guided trials are unlikely to be conducted. 6. Incremental but meaningful contribution: While the absolute contribution of this study may appear incremental relative to prior HER2DX publications, we believe it offers a clinically relevant advance by validating the assay’s utility in a historically understudied population. Our goal is not to generalize beyond HER2DX, but rather to contribute specific evidence that supports individualized treatment decisions in older adults with HER2-positive disease.	
I will comment on specific issues: - There is no data availability statement in the text, and it is a shame that data is not shared with the community. I understand that many times clinical trials decide not to share the data, but at least the statement (and the explanation) should be included in the manuscript. This also makes harder to review the findings of the paper	We thank the reviewer for this important observation. In response, we have added a data availability statement in the manuscript, clarifying the restrictions on data sharing and the conditions under which access may be granted. Specifically, the clinical and genomic data used in this study are subject to intellectual property and licensing restrictions related to the HER2DX assay. Nonetheless, we emphasize that academic access can be granted upon reasonable request, subject to a data transfer agreement and approval by the data custodians. The updated text reads: “The protocol of the N-SAS BC 07 (RESPECT) study is available as a Supplementary file (Supplementary Note 1). The clinical and genomic data generated and analyzed in this study are not publicly available due to intellectual property and licensing restrictions related to the HER2DX assay. However, data can be made available to academic researchers upon reasonable request, subject to a data transfer agreement and approval by the data custodians. Requests should be directed to alprat@clinic.cat.” We hope this addition appropriately addresses the reviewer’s concern.	Page 15, lines 548-553
- The authors perform stratified Cox models using treatment type as a stratification factor. Stratified Cox models assume different baseline hazard functions (for each treatment, in this case) but the same hazard ratio for	We thank the reviewer for this important and constructive comment. The methods used for the primary analysis were not data-driven; they were defined prior to data analysis. As the potential impact of treatment on survival outcomes was uncertain, we decided to adjust for treatment arm in the prognostic analysis of HER2DX. The initial hypothesis was that HER2DX would be equally prognostic across treatment arms (i.e., we assumed the same hazard ratio across treatments), but we wanted to account for potential differences in	Page 6, lines 188-194

all treatments. The authors should explain the reason behind this modelling assumption. Particularly as they later perform tests to see different effects for treatment to test PCR prediction.	baseline hazard between treatments. The use of a stratified Cox model is common in such cases, where the main variable of interest (in this case, HER2DX) is evaluated while adjusting for other variables. Additionally, we also conducted multivariable analyses including treatment as a covariate to confirm that the results were consistent. Stratified Cox models were used in the primary prognostic analysis of the HER2DX risk score in this older population. In contrast, standard Cox models without stratification were used in the exploratory analysis of the predictive value of HER2DX. These two analyses addressed complementary clinical questions, prognostic utility across the cohort and predictive utility for chemotherapy benefit. In sum, treatment was included as a stratification factor when assessing prognostic associations of HER2DX scores, and as an interaction term when assessing predictive value. This modeling strategy has now been explicitly explained in the revised manuscript to avoid confusion.	
- This is a very specific population, as patients are 74 years old on average, followed-up for 9 years, and none of the die because of non-cancer related causes. Unfortunately we don't have any information on censoring, etc but this seems to be very different to what would happen in most countries. Would this limit the findings in the study? Some comment on this would be needed in the text.	We thank the reviewer for raising this important point. We agree that the RESPECT population is unique, with a median age of 74 years, long follow-up, and a remarkably low incidence of non-cancer-related deaths. This likely reflects two factors: (i) the high life expectancy among women in Japan (87.1 years), and (ii) the selection of relatively fit older adults who were eligible for randomization and therefore likely had fewer comorbidities than the general older population. These features may limit the generalizability of our findings to countries with shorter life expectancy or higher competing risks of non-cancer mortality. At the same time, we note that life expectancy continues to increase in many developed countries, and a substantial proportion of individuals >70 years remain in good functional status with limited comorbidities. Moreover, cancer incidence itself increases with age: currently, patients aged ≥ 65 years represent $\sim 50\%$ and those ≥ 80 years represent $\sim 13\%$ of all cancer diagnoses, and these rates are projected to rise to 60% and 21%, respectively, by the mid-21st century. Breast cancer is the most common malignancy among older women, with a peak incidence near age 65 in many countries. Treating this growing population remains a challenge, as increasing frailty and comorbidities are accompanied by greater susceptibility to adverse events, highlighting the importance of treatment strategies that are both effective and tailored to geriatric assessment. These demographic and clinical considerations underscore the broader clinical relevance of evaluating HER2DX in older populations beyond Japan. To address this important point, we have now included the following sentence in the limitations section of the Discussion: “Lastly, the RESPECT trial population was highly specific, with a median patient age of 74 years, 9 years of follow-up, and a remarkably low incidence of non-cancer-related deaths. This likely reflects two factors: (i) the high life expectancy among women in Japan (87.1 years), and (ii) the selection of relatively fit older adults eligible for randomization, who probably had fewer comorbidities than the general older population. These features may limit the	Discussion section, page 16, lines 521-534

	generalizability of our findings to settings with shorter life expectancy or higher competing risks of non-cancer mortality. However, life expectancy continues to increase in many developed countries⁴³, and cancer incidence itself rises steeply with age: currently, patients aged ≥ 65 years account for approximately half of all cancer diagnoses and those ≥ 80 years for $\sim 13\%$, proportions expected to further increase in the coming decades⁴⁴. Breast cancer is the most commonly diagnosed cancer in older women, with a peak incidence near age 65 in many countries⁴⁴. A substantial proportion of these individuals remain in good functional status, underscoring the clinical relevance of our findings and the need for tools such as HER2DX to support tailored treatment decisions in this growing population.” We hope this addresses the reviewer’s concern and provides important context for interpreting our results.	
- I would expect that age would have some effect on survival in these patients, but this has not been tested in table S1	Following the reviewer’s comment, we have performed additional analyses to evaluate the relationship between age, survival outcomes, and HER2DX scores. First, age was evaluated as a continuous and binary variable in univariable models (Tables S1–S4). As expected by the reviewer, age was statistically associated with all survival outcomes. Additionally, to complement the HER2DX analysis and adjust for age, we have added a new section to the manuscript that includes new analyses on the relationship between survival outcomes, age, and HER2DX. Importantly, the association between age and outcomes did not affect the prognostic ability of the HER2DX risk score. When adjusted for age, the HER2DX risk score remained independently associated with both RFS and OS in the overall and node-negative populations. These findings support the robustness of HER2DX’s prognostic utility regardless of age. These findings are now reported in a new section titled "HER2DX Risk Score, Age and Survival Outcomes." “HER2DX Risk Score, Age and Survival Outcomes Age was significantly associated with RFS (HR=1.21; 95%CI 1.07-1.36; p=0.002) and OS (HR=1.26; 95%CI 1.09-1.46, p=0.002) (Tables S1-S2). The age distribution did not differ between patients with HER2DX high- and low-risk scores (Table 1). The median age in patients classified as HER2DX high and low risk score was 74 and 73 years, respectively (p=0.20). After adjusting by age, similar results between HER2DX risk and survival outcomes were observed (RFS; all patients: HR=0.57; 0.27-1.21; node-negative: HR=0.42; 0.17-1.03) and (OS; all patients: HR=0.42; 0.18-0.99; node-negative: HR=0.21; 0.08-0.56).” Additionally, we did not find an association between age and chemotherapy benefit, which is consistent with the findings in the original and full dataset reported by Sawaki et al. JCO 2020 in Table 2 (link). This result is now included in the result section: “HER2DX Risk Score, Age, Hormone Receptor status and Treatment Interaction”	Results section page 9, lines 288-295; Page 10, lines 346-347; Supplementary tables 1-4

	“HER2DX Risk Score, Age, Hormone Receptor status and Treatment Interaction .... Similarly, age (<74 vs ≥74 years) was also not significantly associated with differential benefit from chemotherapy, with interaction p-values of 0.296 for RFS and 0.449 for OS.”	
- Clinical variables are not associated with prognosis, however we see that node status and grade are very different between HER2DX groups. This is a bit surprising, but difficult to understand without looking at the data. Especially as the most predictive module in the score seems to be proliferation. Is this correlated to node and grade? What happens if the use node as continuous instead of categorical?	We appreciate the reviewer’s detailed observations. Regarding nodal status, we confirm that in our dataset, all node-positive patients had pN1 disease (we did not have data on the exact number of positive lymph nodes beyond that). The Kaplan-Meier curves for both OS and RFS show a visual trend toward worse outcomes in patients with pN1 disease compared to pN0, but this difference was not statistically significant (log-rank $p = 0.911$ for OS and $p = 0.723$ for RFS). In addition, we explored whether the biological modules used in the HER2DX Risk Score were associated with nodal status. None of the three modules (proliferation, immune/IgG, and luminal) were significantly associated with nodal status when tested independently ($p = 0.49, 0.51, \text{ and } 0.73$, respectively). This suggests that the biological programs driving the HER2DX Risk Score provide information that is distinct from nodal status, which is why all of these variables were originally included in the final HER2DX Risk Score. Regarding tumor grade, we unfortunately do not have access to this variable in the RESPECT dataset and cannot assess its correlation or prognostic impact.	NA
- The text overstates the results a bit, highlighting the prognostic value of the test. For example, the 10-year RFS has a HR=0.49 with a confidence interval (0.23-1.01), which could not be considered significant by any means.	We agree with the reviewer and have revised the manuscript throughout to better reflect the strength of the evidence. In particular, we have replaced or removed expressions such as “strong,” “robust,” “clinically meaningful,” and “significantly” when describing the prognostic value of HER2DX. Our revised language emphasizes that while HER2DX demonstrated prognostic trends and retained independent associations in multivariable analyses, some of the confidence intervals are wide and the findings should be interpreted with appropriate caution. We have updated the Abstract, Discussion, and Conclusion sections accordingly (please see revised text highlighted in the manuscript).	Page 2, lines 54 and 59; Page 11, line 350; Page 14, line 458
- In particular, the results on pCR prediction are clearly non significant. Only when performing several tests one of them reaches a p-value of 0.045, but Figure 3 clearly shows no effect. This is also overstated in the text, in particular in the conclusions (lines 322-326). The data does not support that statement.	We thank the reviewer for this important and constructive comment. We agree that the interaction p-value of 0.045 for OS in the pCR-high subgroup should not be interpreted as conclusive evidence of a treatment effect, especially in the context of exploratory subgroup analyses and multiple comparisons. However, given the relatively small number of events in the analysis (resulting in low statistical power), it is difficult to detect these differences. While they may serve as hypothesis-generating, they open the door for further exploration of this hypothesis. Accordingly, we have revised the Abstract, Results, and Discussion sections to reflect a more cautious interpretation of the pCR score data and to avoid any overstatement of the findings: In the Abstract, we now state: “The HER2DX pCR score was not prognostic overall. However, chemotherapy was associated with numerically improved survival in the pCR-high group. These findings suggest potential biological	Abstract (page 2, lines 57-59), Results (page 310, lines 322-332), Discussion (page 12, lines 361-381)

differences across HER2+ tumors and may help refine future treatment strategies in older patients.”

In the Results, the relevant paragraph now reads:

“When assessing treatment outcomes within these groups, a non-significant trend toward improved RFS was observed with chemotherapy in the pCR-high group (HR=0.44, 0.14–1.44), with 10-year RFS rates of 88.9% in the chemotherapy and trastuzumab arm and 76.8% in the trastuzumab-alone arm (Figure 3A). Similarly, numerically better OS outcomes were noted in this group (HR=0.23, 95% CI 0.05–1.08), with 10-year OS rates of 94.4% and 77.4%, respectively (Figure 3B). In contrast, no evidence of benefit from chemotherapy was seen in patients classified as pCR-medium/low for either RFS (HR=1.18, 0.45–3.10; 10-year RFS: 70.6% with chemotherapy and trastuzumab vs 76.2% with trastuzumab alone) or OS (HR=1.70, 0.50–5.60; 10-year OS: 76.2% vs 88.6%) (Figure 3C-D). The interaction p-values for RFS and OS were 0.200 and 0.045, respectively (Tables S6-7). These exploratory findings should be interpreted cautiously given the sample size and borderline significance.”

In the Discussion, we now include the following revised paragraphs:

“The HER2DX pCR likelihood score identified a subgroup in which a numerical benefit from chemotherapy was observed. In this analysis, patients classified as pCR-high appeared to derive greater benefit from the addition of chemotherapy, particularly in terms of OS, whereas no such benefit was observed in the pCR-medium/low subgroup. Although the interaction p-value for OS was nominally significant, these findings stem from exploratory subgroup analyses with limited statistical power and should be interpreted with caution.

Interestingly, in Trans-RESPECT, 51.3% of tumors were classified as pCR-high—a proportion notably higher than the ~33% typically observed in neoadjuvant cohorts. A similar pattern was seen in the ATEMPT trial, where 78.6% of patients were pCR-high, suggesting that this distribution may be characteristic of early-stage, clinically low-risk HER2+ tumors treated in adjuvant de-escalation settings. These findings raise the hypothesis that pCR-high biology is enriched in small, low-burden HER2+ cancers, possibly reflecting strong HER2 signaling and immune infiltration even in early-stage disease.

Notably, the pCR and risk scores were uncorrelated in this cohort ($r = -0.009$), confirming that HER2DX’s prognostic and predictive components capture distinct biological dimensions. Importantly, 34% of patients in this cohort were classified as both HER2DX low-risk and pCR-medium/low. This overlap between the prognostic and predictive components of HER2DX suggests that combining these two scores may offer complementary insights when tailoring treatment strategies. In particular, this dual-classification approach could help identify older individuals with HER2-positive early breast cancer who may be appropriate candidates for de-escalated therapies, including the omission of cytotoxic chemotherapy. However, these results should be interpreted with caution given the limited sample size and exploratory nature of the subgroup analyses.”

	We trust that these revisions appropriately temper our interpretation and align with the reviewer's suggestion to avoid overstating the predictive value of the HER2DX pCR score.	
- I don't understand the point of Table S2. The authors are fitting a Cox proportional hazards model, which assumes the same hazard over time. Why the need of splitting the follow-up time and fitting Cox (again) models? From a statistical point of view, this is not a very elegant analysis. It would be better to check the proportional hazards assumption, and if they do not hold, modify the model. Furthermore, the number of tests performed increases to the point where multiple testing correction would be needed.	We thank the reviewer for raising this point and apologize for any confusion caused. The table the reviewer refers to appears to be Table 2, not Table S2, as Table S2 presents a standard multivariable Cox model evaluating clinical-pathological variables. We agree that, from a purely statistical perspective, it makes more sense to check the proportional hazards assumption rather than censoring at different time points. However, from a clinical point of view, there is always uncertainty about whether, in non-metastatic contexts with few events, the results might be influenced by the cohort's follow-up time. In this regard, and in line with the reviewer's comment, we have formally tested the proportional hazards assumption using Schoenfeld residuals. But at the same time, we would like to keep Table 2. The purpose of this table is to provide a sensitivity analysis to explore the consistency of the HER2DX Risk Score's prognostic impact over time in a descriptive and visual manner. This was achieved by censoring patients at each annual timepoint and calculating the hazard ratio between HER2DX risk groups within that truncated follow-up window. The goal was not to derive multiple independent statistical inferences, but rather to illustrate whether the magnitude of the prognostic association was stable over time. The additional sentence has been added: "The proportional hazards assumption was tested and inspected visually using Schoenfeld residuals." We hope this addresses the reviewer's concern.	Table 2 and statistical section
- On the same topic, If the authors are interested in prognosis at specific time points, the authors could treat the event as a binary variable and compute sensitivity/specificity. At least that would add some extra information to the analysis	We appreciate the reviewer's suggestion. Indeed, assessing sensitivity and specificity at specific timepoints can be useful in certain contexts, particularly for binary classifiers. However, the HER2DX Risk Score is a continuous genomic score developed and validated using survival analysis methods, which take into account both the time-to-event nature of the data and censoring. Given the relatively small sample size and the limited number of events in this study, we believe that estimating sensitivity and specificity for relapse-free or overall survival at specific timepoints (e.g., 10 years) could lead to unstable and potentially misleading estimates. In addition, such binary classification would oversimplify the continuous and time-dependent prognostic nature of the HER2DX score. Instead, we chose to present hazard ratios over time (Table 2) as an exploratory way to assess the consistency of the prognostic effect, while preserving the time-to-event framework. As noted above, these results are descriptive and not intended for formal inference. We agree that sensitivity/specificity analyses could provide complementary	NA

	insights in larger datasets with more events and minimal censoring, and we will consider incorporating this approach in future studies.	
- Why are all the hazard ratios for node-negative RFS non significant in Table 3 while in Figure 1C is?	We thank the reviewer for this observation. The difference between Table 3 and Figure 1C is due to both the patient groups analyzed and the type of statistical analysis used. Table 3 includes only patients from the trastuzumab-only arm and shows exploratory time-split analyses at yearly intervals. These analyses exclude patients censored before each timepoint, which reduces the sample size and statistical power, explaining why the hazard ratios are not significant. In contrast, Figure 1C includes all node-negative patients, regardless of treatment arm, and uses the full follow-up period.	NA
- The analysis of the different modules of Her2DX is very interesting, but I would have expected a more thorough analysis. Figure 2A for example, could also reflect output on the vertical axis. Figure 2B is not very informative. Again, there are too many models fitted here (multiple correction testing needed). It would be much better to show Kaplan Meier curves for percentiles to see the prognostic value of each of them. The clinical and Her2 should be also added. Figures 2C and 2D are interesting, but I don't understand why the authors created a new composite score selecting specific parts of the score. The authors need to show that the original score is monotonic with respect to risk. This is the most interesting part of the paper in my opinion, and it could be expanded.	We thank the reviewer for these insightful comments. We agree that the biological decomposition of the HER2DX risk score is a particularly relevant aspect of this study, and we have revised and expanded this section accordingly. First, we have updated Figure 2A to better visualize the biological heterogeneity of the cohort. The heatmap now includes the HER2DX risk classification alongside the percentile scores of the three core biological signatures, proliferation, IGG/immune, and luminal, used in the HER2DX prognostic algorithm. These were selected as they directly contribute to the risk model. The HER2 amplicon signature, while available, is not part of the risk score and was therefore not analyzed in this context. Second, we have now incorporated tumor-infiltrating lymphocytes (TILs) into the analysis, as a biomarker that has been previously associated with prognosis in early-stage HER2-positive breast cancer. TILs showed only a moderate correlation with the HER2DX immune signature (Pearson $r = 0.66$), and in univariable Cox models, TILs, analyzed both as a continuous and dichotomous variable, were not significantly associated with recurrence-free or overall survival. This reinforces the added value of the gene expression-based test over morphology-based biomarkers such as TILs. Third, to improve interpretability, we clarified the rationale behind Figure 2B. While we acknowledge the multiple comparisons involved, these univariable Cox models were not used for formal hypothesis testing, but rather to explore the directionality and consistency of each biological signature's association with long-term outcomes. All signatures showed hazard ratios in the expected direction. In response to the reviewer's suggestion, we have retained Figures 2C and 2D, which show Kaplan–Meier curves based on a composite score ranging from 0 to 3, derived by assigning one point for each favorable biological profile (immune-high, luminal-high, and proliferation-low). These analyses demonstrate a clear monotonic relationship between increasing composite score and improved outcomes, highlighting the additive value of these biological components, which are part of the HER2DX risk score.	Table 1, Figure 2A-E, Tables S1-4, Figure S3, Page 8, lines 281-317

	We believe this biologically driven composite score provides a clear and interpretable demonstration of a monotonic risk gradient based purely on gene expression–derived biology. While the current study focuses on older patients, we note that the monotonic prognostic association of the full HER2DX risk score with survival outcomes has been robustly demonstrated in a recently published patient-level meta-analysis of 2,518 patients across 11 studies (Villacampa et al., Lancet Oncology, 2025). The novelty of the present work lies in showing that the same biological signals underlying HER2DX maintain their prognostic relevance in older patients, a population often underrepresented in clinical trials and biomarker research.	
- Line 226: "Univariate" should be "univariable", as the authors are referring to models with one independent variable	We thank the reviewer for this correction. We have replaced “univariate” with “univariable” throughout the manuscript wherever applicable to accurately reflect the use of statistical models with a single independent variable.	Page 7, line 239; Page 8, line 274; Page 9, line 293; Page 19, line 652
- What treatment effect is supposed to predict the pCR part of HER2DX? Is it trastuzumab or trastuzumab+chemo? The trial did not find any difference in treatment.	We thank the reviewer for this insightful question. The HER2DX pCR score is designed to estimate the intrinsic likelihood of achieving a pCR following neoadjuvant trastuzumab-based chemotherapy in early-stage HER2+ breast cancer. The score captures tumor-intrinsic biological features, such as proliferation, luminal, immune infiltration, and HER2 signaling, all of which have been associated with response to HER2-targeted therapy and chemotherapy (see our recent review Waks et al. Nature Rev Clin Oncol 2024). Thus, the HER2DX pCR score is not intended to isolate the effect of trastuzumab alone versus trastuzumab plus chemotherapy, but most likely reflects the combined treatment effect, as typically delivered in clinical practice. This interpretation is supported by prior studies. In the patient-level meta-analysis of 765 patients across seven neoadjuvant trials (Villacampa et al., Ann Oncol, 2023), tumors classified as pCR-high by HER2DX achieved pCR rates of 80–90% when treated with a single taxane and dual HER2 blockade. In contrast, in the PHERGain trial (Llombart-Cussac et al., Clin Cancer Res, 2024), the same tumors had pCR rates of only 44.6% when treated with trastuzumab and pertuzumab alone without chemotherapy, despite patient selection based on early PET response and the addition of endocrine therapy in hormone receptor–positive disease. These findings reinforce the idea that the pCR score captures sensitivity to combined chemotherapy plus HER2-targeted regimens. In the RESPECT study, the score was used exploratorily to evaluate whether tumors predicted to have a high likelihood of pCR (i.e., pCR-high) were also more likely to benefit from the addition of chemotherapy. As shown in Figure 3A–D, patients classified as pCR-high demonstrated a non-significant trend toward improved RFS and OS with chemotherapy plus trastuzumab, whereas no such benefit was seen in the pCR-medium/low group. The interaction p-value for OS was 0.045, but given the sample size and exploratory nature of the analysis, these findings should be interpreted with caution. We have revised the Discussion section to clarify the biological rationale and to reference these findings.	Page 13, lines 419-439

	Revised paragraph in Discussion section: “The HER2DX pCR likelihood score is a genomic predictor developed to estimate the probability of achieving a pCR following neoadjuvant trastuzumab-based chemotherapy in early-stage HER2-positive breast cancer. It reflects tumor-intrinsic features, such as proliferation, luminal, immune infiltration, and HER2 signaling, that are associated with chemosensitivity and response to HER2-targeted agents. In this study, the pCR score identified a subgroup of patients (pCR-high) in which a numerical benefit from chemotherapy was observed. Specifically, patients classified as pCR-high appeared to derive greater benefit from the addition of chemotherapy to trastuzumab, particularly in terms of OS, whereas no such benefit was seen in the pCR-medium/low group. These findings are biologically plausible, as tumors with a high HER2DX pCR score often display aggressive but chemo-sensitive features, including high proliferation, strong HER2 signaling, and robust immune infiltration. Supporting this, a recent patient-level meta-analysis including 765 patients reported pCR rates of 80–90% in pCR-high tumors treated with a single taxane plus dual HER2 blockade. In contrast, in the PHERGain trial, the same pCR-high tumors achieved only 44.6% pCR when treated with trastuzumab and pertuzumab alone without chemotherapy, despite PET-based patient selection and the use of endocrine therapy in hormone receptor–positive disease. This suggests that the full therapeutic effect in these tumors relies on the combination of chemotherapy and HER2-targeted therapy. Although the interaction p-value for OS in RESPECT was nominally significant, these analyses were exploratory, based on small subgroups, and should be interpreted with caution.”	
- Can you call "high risk" a group with 75% survival after 5 years in a population that is 74 years old? Would this term be communicated to the patient?	We thank the reviewer for this thoughtful and important comment. We fully agree that terms such as “high risk” must be used with care, particularly in older populations where absolute survival may remain relatively high and competing risks of death are present. In the context of HER2DX, the term “risk” refers specifically to the risk of distant recurrence and breast cancer–specific death, not to all-cause mortality. The HER2DX risk score was developed to estimate disease-specific prognosis and is independent of patient age, comorbidities, or competing risks. The label “high risk” is a relative classification within the HER2DX framework. In the ShortHER training dataset, the risk stratification cutoff was predefined to ensure that the low-risk group maintained >90% distant recurrence-free survival at 3, 5, and 7 years, regardless of trastuzumab duration. Patients classified as high-risk fall outside this threshold and have a comparatively higher risk of recurrence, even if their absolute prognosis remains favorable in certain populations, such as older adults. We agree that while this terminology is appropriate in the context of risk stratification and treatment decision-making, it may not be optimal for direct communication with patients. In clinical practice, we support individualized, patient-centered communication that	Page 5, line 164-167

	incorporates absolute risk estimates and considers the broader clinical context. To clarify the basis of the HER2DX classification, we have added the following sentence to the Methods section: “In the ShortHER training dataset, the risk stratification cutoff was predefined to ensure that the low-risk group maintained >90% distant recurrence-free survival at 3, 5, and 7 years. Patients classified as high-risk fall outside this threshold and have a comparatively higher risk of recurrence.”																									
- Any reason why low/medium risk are merged? Shouldn't be better to check if there is a monotonic effect in the prediction score in Figure S3?	We appreciate the reviewer’s suggestion. In the main analysis, we combined the HER2DX pCR score low and medium groups to increase interpretability and statistical power, as this grouping has been used in prior HER2DX publications and reflects tumors with lower expected chemosensitivity. To address the reviewer’s comment, we performed an additional exploratory subgroup analysis separating the pCR score into low, medium, and high categories. As shown in the newly added Table S7, the estimated hazard ratio for chemotherapy (vs. no chemotherapy) on overall survival was lowest in the pCR-high group (HR = 0.23), suggesting a stronger association with benefit. In contrast, the pCR-medium and pCR-low groups had HRs of 1.37 and 2.12, respectively, indicating little or no observed benefit. Table S7. Association between chemotherapy addition and overall survival by HER2DX pCR score subgroup.    HER2DX pCR-group Low Medium High     N 42 33 79   Hazard Ratio (chemotherapy vs no chemotherapy) 2.12 1.37 0.23   95% CI (lower) 0.21 0.33 0.05   95% CI (upper) 21.25 5.75 1.08   P-value (log-rank test) 0.5 0.7 0.04   	HER2DX pCR-group	Low	Medium	High	N	42	33	79	Hazard Ratio (chemotherapy vs no chemotherapy)	2.12	1.37	0.23	95% CI (lower)	0.21	0.33	0.05	95% CI (upper)	21.25	5.75	1.08	P-value (log-rank test)	0.5	0.7	0.04	Table S7
HER2DX pCR-group	Low	Medium	High																							
N	42	33	79																							
Hazard Ratio (chemotherapy vs no chemotherapy)	2.12	1.37	0.23																							
95% CI (lower)	0.21	0.33	0.05																							
95% CI (upper)	21.25	5.75	1.08																							
P-value (log-rank test)	0.5	0.7	0.04																							
Minor comment: - Line 182: "For data analysis, the HER2DX pCR group was low and medium was pooled in..." It should say "For data analysis, the HER2DX pCR group *that* was low and medium was pooled in..."	We thank the reviewer for the suggestion. The sentence has been revised as follows for clarity: “For data analysis, the HER2DX pCR group that was low and medium was pooled into a single category.”	Page 6, line 204-205																								
In summary, although I agree that this is a very interesting subpopulation to test, there is no access to the data, limiting the interest to the scientific community, and the	We thank the reviewer for this final comment and appreciate the opportunity to clarify. While HER2DX has indeed been validated in larger datasets, this study represents the first evaluation of its prognostic and predictive performance in a randomized trial exclusively enrolling older patients (aged 70–80 years) with HER2-positive early breast cancer. Older adults remain underrepresented in clinical trials and are often excluded from studies informing biomarker	NA																								

performance of the test has been already quantified in several other studies in larger cohorts. The fact that the effects here are smaller, that many tests have been performed without control for multiple testing and the overstatement of the findings limit the interest of this study in my opinion.	development and treatment de-escalation strategies. Our study directly addresses this evidence gap and provides novel data to support a more personalized approach in this understudied population. In addition to demonstrating the long-term prognostic value of HER2DX in this setting, our work shows that the same biological mechanisms driving outcome and treatment response in younger patients are also operative in older individuals. These findings reinforce the relevance of biology-based risk stratification irrespective of age and have potential clinical implications for optimizing treatment decisions in older patients—particularly the use or omission of chemotherapy. While subgroup analyses were exploratory, they are biologically plausible and consistent with prior studies. We have revised the manuscript to carefully qualify these findings and avoid overstatements. Regarding the reviewer’s concern about data access, we have now included a Data Availability statement in the manuscript. Although the full dataset is not publicly available due to licensing and intellectual property restrictions, access can be granted to academic researchers upon reasonable request and pending appropriate agreements. We believe this ensures scientific transparency while respecting the constraints associated with a regulated genomic assay. We hope these clarifications address the reviewer’s concerns and highlight the scientific and clinical value of the work.	
Reviewer #3:		
This paper describes a pre planned investigation of the HER2DX genomic assay in a subgroup of patients within the randomised RESPECT Trial (Trans-RESPECT). The HER2DX score, and its individual component, was independently prognostic of survival. While its pCR score was not significantly prognostic, chemotherapy was of benefit in the high risk group only. Data provides further evidence to support the use of this assay for individual patient management. The study is well designed, the aims are clear and the analyses are relevant. The results are well represented and supported by tables and graphs. The conclusion	We sincerely thank the reviewer for the positive assessment of our study. We are pleased that the reviewer found the design, analysis, and presentation of the results to be appropriate and aligned with the study objectives. We also appreciate the recognition of the relevance of our findings in supporting the use of HER2DX for individual patient management, particularly in older adults with HER2-positive early breast cancer.	NA

follows well from the results.		
There are few queries and points that need clarification to improve readability and further strengthen the manuscript: The authors state in the introduction: “However, its utility in older patients or those receiving trastuzumab monotherapy has not been previously explored”. Please explain current utility in the standard population. Despite significant evidence, this has not been incorporated into routine practice and clinical guidelines. Please explain and provide recommendations for its use.	We appreciate this thoughtful suggestion. We have now expanded the Introduction to better contextualize the current utility of HER2DX in early-stage HER2+ breast cancer. Specifically, we explain that HER2DX has demonstrated consistent prognostic and predictive performance across multiple clinical trial cohorts, including APT and ATEMPT, and that a recent prospective real-world study showed HER2DX results led to treatment adjustments in nearly half of patients—most often via reduced treatment intensity—while maintaining confidence in clinical decision-making. We also clarify that HER2DX is incorporated into the Spanish national breast cancer guidelines (SEOM, 2023) and was discussed at the 19th St. Gallen Breast Cancer Conference in 2025. Additionally, we highlight that HER2DX is used to guide decision-making across various clinical scenarios, including chemotherapy intensity, use of neoadjuvant therapy, and HER2 status confirmation. These revisions help underscore the relevance of HER2DX in standard practice and frame the novel contribution of our study in assessing its utility specifically among older patients treated with trastuzumab with or without chemotherapy.	Page 2, lines 103-116
All cause survival was analysed. Did the authors look at breast specific survival?	We thank the reviewer for this important question. Breast cancer–specific survival data were not formally collected in the RESPECT trial; therefore, our analyses were based on OS, which was the primary endpoint of the original trial. However, we note that in this dataset, all deaths occurred following a documented breast cancer recurrence, indicating that OS largely reflects breast cancer-related mortality in this cohort. This strengthens the relevance of OS as an endpoint to assess the prognostic value of HER2DX in this older population.	NA
For the 6 cases that were not profiled by HER2Dx, please provide the reason(s) (e.g amount of tissue, quality of mRNA...etc).	We thank the reviewer for this comment. The information was included in the original submission as part of Figure S1, but we have now integrated the specific details into the main text for clarity. In brief, 172 FFPE tumor blocks were retrieved from local sites. Of these, 166 (96.5%) met the minimum tumor requirements, 163 (98.2%) met the minimum RNA requirements, and 154 (94.5%) yielded successful HER2DX results, corresponding to an overall profiling success rate of 89.5% from block retrieval to final results. The six cases that did not produce HER2DX results despite meeting RNA quantity thresholds failed due to insufficient RNA quality. The revised text in the Results – Patient Characteristics section now reads: “A total of 154 out of 266 eligible patients (58.0%) from the RESPECT trial had tumors successfully profiled with the HER2DX test and were included in this analysis. This subset reflects the 172 FFPE tumor blocks retrieved from local sites that agreed to participate in the translational study, of which 166 (96.5%) met the minimum tumor requirements, 163 (98.2%) met the minimum RNA requirements, and 154 (94.5%) yielded successful HER2DX results (Figure S1). The overall profiling success rate from tumor block retrieval to HER2DX results was 89.5%.”	Page 7, lines 226-230

How did the pCR score obtained by HER2Dx correlate with the pathological complete response assessed by pathologists? Was the latter prognostic for survival in the study cohort, considering the HER2DX score was not?	We agree this is an important point. However, in the RESPECT trial all patients underwent primary surgery before adjuvant therapy; therefore, no pathologic complete response (pCR) assessment was performed, and no correlation between the HER2DX pCR score and pathologist-assessed pCR could be evaluated in this cohort. In the present study, the pCR score was explored only as a biological signature and as a potential prognostic marker. As stated in the Results section, when evaluated as a prognostic marker, the pCR score was not significantly associated with either RFS or OS (Figures S5–S6).	NA												
What was the relation between HER2DX Her2 ERBB2 status (low, medium, high) and HER2 clinical status by immunohistochemistry/ISH ?	We appreciate the reviewer’s question. Unfortunately, HER2 IHC scores or ISH results beyond the binary positive/negative classification were not available for the RESPECT trial samples, so we could not directly evaluate the association between HER2DX ERBB2 score categories and IHC/ISH levels in this cohort. However, the relationship between HER2DX ERBB2 score and clinical HER2 status has been previously characterized in detail in large independent datasets. In those studies, the HER2DX ERBB2 score, reported on a scale from 1 to 99 with predefined cut-offs distinguishing ERBB2-low (1–32), ERBB2-medium (33–50), and ERBB2-high (51–99), showed excellent performance in predicting clinical HER2 status, with an AUC of 0.96–0.98 across >1,400 tumors. Importantly, none of the HER2-negative cases (IHC 0/1+ or 2+/ISH-negative) were categorized as ERBB2-medium/high. In the present study, most tumors (79.9%) were classified as ERBB2-high/medium, which is in line with previous reports. Table 1 in the revised manuscript now shows the distribution of ERBB2 score categories for the overall cohort. For the reviewer’s information, the relationship between ERBB2 score and HR status in the RESPECT cohort is shown below:    HR status ERBB2-low ERBB2-medium ERBB2-high     HR-positive 19 (24.1%) 20 (25.3%) 29 (36.7%)   HR-negative 12 (15.2%) 12 (15.2%) 62 (78.5%)    As expected, HR-negative tumors were more frequently ERBB2-high compared with HR-positive tumors, consistent with prior findings (e.g., Prat et al., EBioMedicine 2022).	HR status	ERBB2-low	ERBB2-medium	ERBB2-high	HR-positive	19 (24.1%)	20 (25.3%)	29 (36.7%)	HR-negative	12 (15.2%)	12 (15.2%)	62 (78.5%)	Table 1
HR status	ERBB2-low	ERBB2-medium	ERBB2-high											
HR-positive	19 (24.1%)	20 (25.3%)	29 (36.7%)											
HR-negative	12 (15.2%)	12 (15.2%)	62 (78.5%)											
It appears that the findings were independent of the hormone receptor status. Perhaps this should be stated/emphasised in the results and the discussion as this is important clinically.	We thank the reviewer for this important comment. Hormone receptor status was not significantly associated with RFS or OS in univariable analyses (Tables S1–S4), and no interaction was observed between hormone receptor status and treatment benefit for either RFS or OS (new data now included in the Results section). We have also updated Figure 2A to show the distribution of HER2DX scores by hormone receptor status. In addition, we have added the following paragraph to the Discussion: “In this study, hormone receptor status was not significantly associated with long-term outcomes, and no interaction was observed between hormone receptor status and treatment. Furthermore, HER2DX risk scores and underlying biological signatures were distributed across both hormone receptor-positive and -negative tumors, indicating that the assay captures biological information that	Figure 2A; Page 10, lines 335-355; Page 12, lines 409-417												

	is not merely a reflection of hormone receptor expression. These results align with (1) prior reports showing inconsistent associations between hormone receptor status and prognosis across studies, and (2) evidence that genomic stratification by HER2DX provides complementary prognostic information to standard clinicopathologic variables, including hormone receptor status, in early-stage HER2+ breast cancer.”	
The conclusions are rather generic. What is the clinical implication of this study? Do authors believe there is now enough evidence to adopt the test for individualised patient management in routine practice?	We appreciate the reviewer’s request for clarification. We have revised the conclusions to better reflect the specific contribution of this study while avoiding overstatement of clinical utility, in line with Reviewer 2’s feedback. The revised text now highlights that this is the first study to evaluate HER2DX in a randomized trial exclusively enrolling older patients with HER2+ early breast cancer, including a trastuzumab monotherapy arm. Our findings confirm the prognostic value of the HER2DX risk score in this underrepresented population and suggest a potential role in refining treatment decisions in older adults, particularly regarding chemotherapy use.	NA
Reviewer #4:		
This retrospective study examines the correlations of HER2DX gene expression analysis with clinical outcomes of patients in the RESPECT trial. It is worth mentioning (Introduction) the rationale or premise of the RESPECT study and briefly explain why the study population (elderly) received breast surgery upfront, when standard practice would be to offer neoadjuvant systemic treatment. Please discuss not giving Pertuzumab to the node-positive population, and therefore the impact on survival analysis.	We thank the reviewer for this comment. We have addressed this point in the Discussion section of the revised manuscript to provide context on the treatment landscape and the rationale for the patient population in RESPECT: “Neoadjuvant systemic therapy is considered the current standard of care for patients with stage II–III HER2+ breast cancer, as it provides prognostic information through assessment of pCR and enables tailoring of adjuvant therapy. In the RESPECT trial, approximately 43.0% of patients had stage I disease, for which neoadjuvant therapy is not standard, while 57.0% had stage II–III disease (41.5% stage IIA, 13.3% stage IIB, and 1.1% stage IIIA), which could have been candidates for neoadjuvant therapy if primary surgery had not been performed. However, neoadjuvant therapy in this setting typically consists of multi-agent chemotherapy regimens such as AC-T or docetaxel–carboplatin, which have substantial toxicity, and, if pCR is not achieved, these patients are subsequently exposed to adjuvant T-DM1. In contrast, in RESPECT, no patient received these regimens. At the time the RESPECT trial was initiated, residual disease–guided treatment following neoadjuvant chemotherapy was not recommended. The trial also pre-dated the approval of neoadjuvant pertuzumab and the results of the APHINITY trial, which demonstrated an overall survival benefit with the addition of pertuzumab in node-positive disease. Nonetheless, the proportion of patients with node-positive disease in RESPECT was low (16%).” This addition clarifies the historical context of the RESPECT trial and explains why primary surgery was performed in this elderly population despite current trends toward neoadjuvant therapy in stage II–III disease.	Page 14, lines 460-474
In usual clinical practice, is it feasible for the diagnostic biopsy alone (enough tissue?) to be used for HER2DX gene expression analysis? This is with respect to giving	We agree with the reviewer that feasibility in the diagnostic biopsy setting is important for the potential use of HER2DX to guide neoadjuvant treatment decisions. The assay has been shown in its analytical validation study, as well as in multiple clinical validation datasets, to be reliably performed on RNA extracted from standard FFPE core needle biopsies obtained at diagnosis, without the need for	Page 14, lines 484-486

neoadjuvant treatment. Please mention in Discussion.	additional tissue. We have now clarified this point in the discussion by adding the following sentence: "Importantly, HER2DX can be reliably performed using RNA extracted from standard FFPE core needle biopsies obtained at diagnosis, without the need for additional tissue, enabling its integration into pre-treatment decision-making."	
--	---	--

3

Reviewer comments	Author response and changes made	Page number in revised, tracked paper where the change can be found
Reviewer #1:		
I agree this manuscript is acceptable for publication as it is currently written. The authors have answer my previous criticism with the following statement in the discussion section. Therefore, our findings cannot be used to infer the benefit of any specific chemotherapy backbone. Finally, our findings are specific to the HER2DX assay and cannot be generalized to other commercially available genomic tests such as Oncotype DX, MammaPrint, EndoPredict, or Prosigna, which differ in development context, gene content, and biological focus⁴³. In addition, HER2DX was designed specifically for HER2+ early breast cancer and uniquely incorporates an immune-related gene signature, which is particularly relevant in predicting prognosis and response to HER2-targeted therapies^{15, 17, 44}. Future comparative studies between HER2DX and other genomic assays in HER2+ populations may help clarify differences in clinical utility and biological coverage across platforms. Congratulations on an excellent study and contributing to the knowledge of treatment for HER-2+ breast cancers.	We thank the reviewer for his/her comment.	NA
Reviewer #2:		
I thank the authors for the additional analysis and rewriting, however I'm not sure I have received the current version of the manuscript. I don't see any highlighted text in the pdf, and some changes are not present. For example, in one of the responses they claim	We apologize for the confusion. The previously submitted clean and tracked-changes versions are correct.	NA

In the Abstract, we now state: “The HER2DX pCR score was not prognostic overall. However, chemotherapy was associated with numerically improved survival in the pCR-high group. These findings suggest potential biological differences across HER2+ tumors and may help refine future treatment strategies in older patients.” but the abstract reads "While the HER2DX pCR score was not prognostic overall, exploratory subgroup analyses suggested a potential survival benefit from chemotherapy in the pCR-high group. HER2DX offers prognostic value and may guide chemotherapy use in older patients with HER2-positive early breast cancer." Am I reading the right version?		
Reviewer #3:		
I thank the authors for satisfactorily addressing my comments.	Thank you very much.	NA
Reviewer #4:		
All the points I raised in the previous review have been addressed satisfactorily.	Thank you very much.	NA